# Measuring the Service Quality of Fresh Food Delivery Platforms: Development and Validation of the "Food PlatQual" Scale

**Jee-Won Kang and Young Namkung \***

College of Hotel & Tourism Management, Kyung Hee University, 26 Kyungheedae-ro, Dongdaemun-gu, Seoul 02447, Korea; jwjw0226@khu.ac.kr
\* Correspondence: ynamkung@khu.ac.kr

**Abstract:** This research conducted three studies to develop a scale for measuring the service quality of fresh food delivery platforms. In Study 1, the scale development stage, a total of 55 items were generated via literature reviews, text mining, and expert interviews. In Study 2, the preliminary assessment stage, the first consumer survey ($n = 550$) was conducted to purify and refine the items derived from Study 1 using exploratory factor analysis (EFA) and confirmatory factor analysis (CFA). Lastly, in Study 3, the second consumer survey ($n = 570$) was carried out to validate the scale using CFA. Ultimately, Food PlatQual scale, consisting of 25 items with seven dimensions: "information quality", "price", "product assortment", "problem resolution", "delivery quality", "ease of use", and "trendiness". The current study is expected to offer a theoretical basis for future research as well as offer useful managerial implications for sustainable fresh food delivery platform services.

**Keywords:** fresh food delivery platform; service quality; scale development; scale validation

## 1. Introduction

Advanced mobile technology allows consumers to shop on demand and buy goods anywhere or anytime with a mobile connection [1]. Online sales of non-perishable products such as clothes, books, and household items are now extending to the fresh food sector as well. In particular, due to improvements of cold chain logistic technology, consumers can also order fresh food online and receive it the next day [2]. The outbreak of COVID-19 has dramatically accelerated the shift toward a larger online shopping environment and triggered changes in consumers' food purchasing behaviors [3].

Due to time savings and convenience, several experts predict that the growth of the fresh food delivery platform business will continue even after pandemic is over [4–6]. Amazon's global sales in the food category are predicted to increase to $26.7 billion by 2026 from $14.5 billion in 2021. Amazon.com has been investing in its online grocery business, and the Amazon Fresh (Seattle, MA, USA) service has been expanding in international markets as well [4]. Further, according to the report by McKinsey and Company (2021) [5], approximately 70% of respondents' preferences for fresh food shopping online changed after the COVID 19 outbreak. Therefore, the fresh food delivery platform would be a new business model and distribution channel to food service industry, and investigation of platform service would be necessary to promote sustainability in the foodservice industry sector.

This study targets South Korea, an advanced IT country with an internet usage rate of approximately 92% [7]. In Korea, online shopping has already become a daily life, and the online fresh food purchase rate is also high. Online shopping for fresh food market in Korea stood at $2.02 billion in 2020 and is forecasted to grow to $10 billion in 2023 [8]. Therefore, various and practical implications can be obtained by conducting research on fresh food delivery platforms in Korea.

With the growth of the fresh food delivery platform business, the service companies have faced intense competition. Thus, the importance of service quality management for customer satisfaction and sustainable growth is increasing [9]. Service quality of fresh food delivery platform could be defined as the degree of difference between the service that consumers expect from the fresh food delivery platform and consumer's perceived performance that they experience when purchasing items through services [10]. Service quality is closely related to the success of business and customer satisfaction [11]. For successful business, companies should identify which service quality attributes their services possess and then determine how to manage these attributes to meet consumers' expectations [12]. Unfortunately, the service quality dimensions and attributes of fresh food delivery platforms have not been thoroughly investigated, and service quality measure has not been developed through a systematic process. Thus, the majority of research of online fresh food purchasing behavior derived their service quality factors by referring to previous studies in similar fields, such as online shopping or other types of online services, rather than applying systematically developed measurement items that reflect the unique features of fresh food delivery platforms. In contrast with manufactures, fresh food and food products are in the nature of perishable and temperature sensitive items [13]. Furthermore, fresh food shopping is more frequent activity than shopping for other product category and a regular and routine tasks in the life [14]. Therefore, the factors such as delivery service, temperature-related factors, easy order process, and easy return should be reflected to consist of service quality dimensions of fresh food delivery platform. An industry-specific instrument that considers the unique characteristics of these market offerings should be developed to evaluate service quality and obtain clearer insights and implications.

Therefore, this study aimed to fill this critical research gap by developing the scale to assess the service quality of fresh food delivery platforms. In this study, we defined the fresh food delivery platform as an online commerce that sells a variety of food products including fresh food and delivers them to consumer's home. This research employed the systematic process of instrument development outlined by Churchill (1979) [15]. Specifically, this study applied a more recent research technique, big data analysis, along with conventional scale development methods. Using big data, this study intended to reveal how consumers perceive fresh food delivery platforms in their daily lives and then reflect these results in the item generation phase. The vast amount of data that accumulates over a long period of time highlights additional service attributes that cannot be obtained through traditional methods (e.g., interview or survey) and provides objective evidence for developing items related to fresh food delivery platform services.

Several steps were taken to develop an instrument for evaluating fresh food delivery platform service quality. First, previous studies of online food purchasing were reviewed. Second, features of fresh food delivery platforms were extracted from big data using text mining and expert interviews. Third, a consumer survey was conducted to refine and purify the items. Fourth, a second consumer survey was implemented to validate the purified items. The present study provides a theoretical foundation for future research regarding scale development and service quality. Further, the Food PlatQual scale developed in this study has practical implications for online food sales businesses.

## 2. Theoretical Background

### 2.1. Fresh Food Delivery Platform

A fresh food delivery platform is an e-commerce specializing in the fresh food category [16]. This platform sells food products, fresh foods (e.g., vegetables, fruits, meat, and fish), refrigerated and frozen foods, and other groceries, and offers next-day door to door delivery service [8]. Many different types of online food sales channels exist, and the services that are offered differ from country to country. Fresh food delivery platforms can broadly be divided into two types: (1) where offline stores expand their business scope online (e.g., Amazon Fresh or Walmart (Bentonville, AR, USA)) or (2) where food is sold only online without any offline stores (e.g., fresh direct). More specifically, these platforms

have been classified as follows: an aggregator model that acts such as an intermediary and only provides an online space for selling products; a single store model in which a single store exists; a store-pick model in which consumers order online and pick up goods at an offline store; and a hybrid model that combines two or more of the previous types [17]. Since there are so many different categorizations, this study defined the fresh food delivery platform broadly as an online based single store that delivers food, including fresh food (e.g., vegetables, fruits, meat, fish, etc.), the same day or the next day, rather than selecting one specific platform or type [8,16]. Since the pandemic began, face-to-face transactions have been restricted and consumers in many countries have started using fresh food delivery platforms as a channel for purchasing food [3,18]. This new food shopping method is emerging as a competitive sustainable advantage in the business environment, and consumers are becoming familiar with the online fresh food buying experience [19].

Several recent studies have investigated how consumers' food purchasing behaviors are changing due to COVID-19. Chang and Meyerhoefer (2020) [18] found that COVID-19 facilitated online food sales by 5.7% and increased the number of users by 4.9%. Further, when the media covered the pandemic, shopping rates were affected. It can be inferred that people are worried about infection and prefer to avoid face-to–face interactions during the pandemic. Ali et al. (2020) [20] examined the effect of COVID-19 on consumers' adoption intentions towards online food delivery services applying the theory of technology readiness (TR). Li et al. (2020) [3] and Alaimo et al. (2020) [19] acknowledged that people are increasingly familiar with buying food online. Specifically, consumers with higher education are more likely to perceive online channels to be easy to use and be satisfied with them.

Some studies have tried to determine which characteristics influence consumer satisfaction and intentions to buy food online. Dominici et al. (2021) [21] explored the relationship between consumers' socio-demographic characteristics and online food purchases. Women, who have traditionally taken care of household duties, such as purchasing household appliances, are more likely to buy groceries online. Further, younger consumers who are more technology-oriented tend to adopt online channels to buy food. However, household size did not have a significant relationship with the tendency to buy groceries online. Piroth et al. (2019) [22] investigated consumers' willingness to buy groceries online using the theory of planned behavior. Their findings indicated that a consumer's peer group influences their perceptions of buying services and intentions to use them. Further, they revealed that this influence on willingness to buy differs according to prior experience.

### 2.2. Dimensionality of Fresh Food Delivery Platforms

With the growth of fresh food delivery platforms in recent years, researchers have become increasingly interested in studying which attributes affect consumers' online food buying behaviors. Previous research indicated a number of service quality attributes for fresh food delivery platforms. Fresh food delivery platforms offer not only products that consumer needs but also information [23]. Information quality is a main service factor that has been suggested in the field of online shopping because it reduces consumers' anxiety about products and satisfies their intellectual needs [24–26]. According to Lee (2016) [27], information quality significantly affected customer trust and satisfaction when using online markets to buy fresh food. Further, Kim and Kim (2018) [23] observed that consumers had indirect experiences by learning about new trends, seasonal products, various food brands, varieties, and recipes. Appropriate Price is a marketing mix that influences the consumer decision making process in shopping more generally, as well as the online food sector [28]. Whether or not a price is fair is directly associated with consumers' feelings and shopping experiences [29]. High prices hinder consumers' purchases [23], and a reasonable price policy is essential to attract potential consumers and maintain existing consumers. In terms of product assortment, Park and Park (2018) [30] found that consumers who use fresh food delivery platforms for psychological satisfaction prefer to purchase various flavors, sizes, and ingredients. Singh (2019) [31] also noted that consumers perceive online

platforms to have a wider assortment of products than offline stores. In addition, some consumers cited the availability of smaller packaged products as an advantage of fresh food delivery platforms [24]. Problem resolution, specifically easy refunds and returns [24,25,32], is important to attract potential consumers. Easy processes for refunds, exchanges, and cancellation are prime requirements for fresh food delivery platforms since consumers usually have a high degree of uncertainty about the products [31].

According to Kim and Kim's (2019) [24] research, consumers who do not purchase fresh food online reported that the difficulty of obtaining a refund was one reason they do not use such services. Delivery quality was discussed in much of the prior research [23–25,27,28,33]. Lee (2018) [25] examined the period of delivery, delivery costs, packaging, and expertise as components of delivery services. Wilson-Jeanselme and Reynolds (2006) [34] stressed that a delivery must be completed on time as promised. Park et al. (2019) [32] addressed that fresh food delivery platforms should pay attention to the delivery service so that products are not damaged and product quality is maintained. The ease and convenience of ordering, payment, navigation, and searching when purchasing food online has been mentioned in many studies and is generally emphasized as a component of quality for various media technologies [23,24,26,32,33]. The current study also examined whether trendiness is a factor that can be considered a service attribute of fresh food delivery platforms. Although trendiness is not a variable that has been frequently discussed in previous research, it has been mentioned in few studies. Martín et al. (2019) [35] asserted that the advantages of online food shopping include modernity and the possibility of trying new products. Pahor et al. (2018) [36] implied the importance of offering new arrivals and trendy products to trigger the impulsiveness of consumers. In addition, Kim and Kim (2018) [23] identified that when consumers use fresh food delivery platforms, they have an emotional experience, are aware of trends, and expect it to change their lifestyle. Since fresh food is a product that easily deteriorates due to time and temperature, product quality has been suggested as a factor that affects consumer purchases [25,37]. Additionally, financial benefits (e.g., discounts and free gifts) [25,28,30,34] and website design [23,27,28,33] were suggested as factors in the literature.

Many researchers have investigated service quality attributes and suggested theoretical models to examine the relationship between these attributes and various behavioral variables [23–26,30,33,34]. However, due to the absence of a scale that specifically measures the service quality of fresh food delivery platforms, most previous studies selected attributes of online food buying based on the researcher's subjective judgement or generic service quality properties of online shopping without considering the unique features of products that are perishable. The more specific the measurement items are in a service quality scale and the more suitable they are to a manager's own contextual situation, the more useful the information will be and the more remarkable insights can be gained [38]. Thus, this research attempted to fill this gap by identifying service attributes and developing a fresh food delivery platform service quality scale—Food PlatQual.

## 3. Overview of the Phases

The service quality scale for fresh food delivery platforms was developed following previous scale development literature [17,39,40]. Development of Food PlatQual included three phases: (1) scale development, (2) preliminary assessment, and (3) scale validation. This study developed potential items based on three sources: research literature, big data, and expert interviews. First, previous literature related to online food purchase was reviewed. Subsequently, big data analysis was performed to ensure that consumers' broad opinions regarding fresh food delivery platforms were reflected in the development of the scale. Expert's interview was carried out to examine the initial pool of items. Then, an online survey was conducted to purify the scales and assess the initial validity. Finally, a second online survey was carried out to validate the scales and test the nomological validity. Figure 1 shows the overall process of developing the Food PlatQual scale.

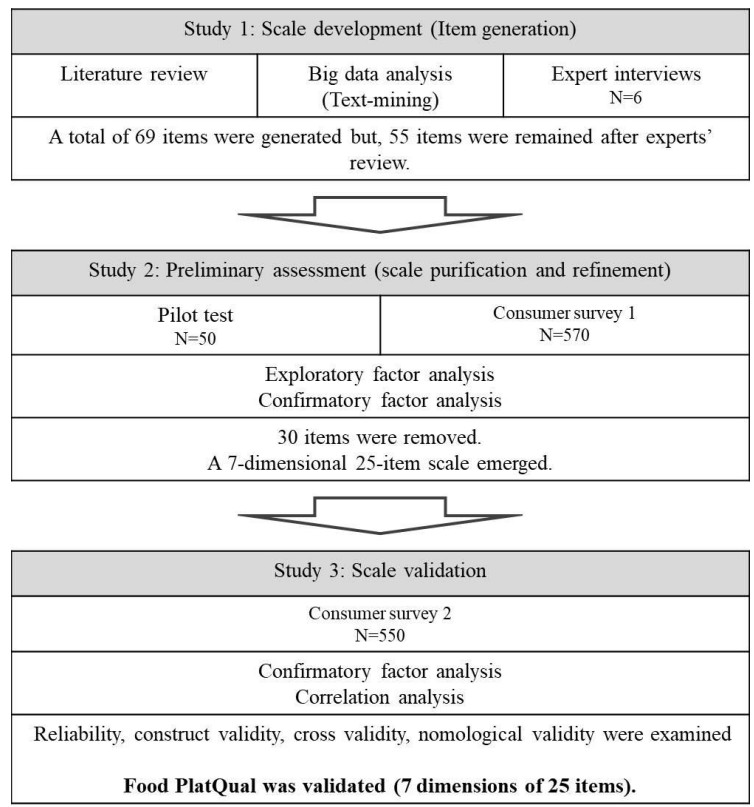

**Figure 1.** Scale development process.

## 4. Scale Development of the Fresh Food Delivery Platform Service Quality

*4.1. Study 1: Scale Development (Item Generation)*

The purpose of the Study 1 was to identify the elements that constitute the service quality of fresh food delivery platforms. Based on the existing literature, the data from text-mining, and expert interviews, as many items as possible were generated that represent service quality.

### 4.1.1. Step 1: Literature Review

The first step in scale development was to review existing literature and figure out various service quality dimensions of the fresh food delivery platform and generate items. Prior studies were reviewed using keywords such as online fresh food purchase, online food shopping, online grocery shopping, and etc. In order to identify what service quality attributes the fresh food delivery platform should have, previous literature related with consumer behavior studies in the field of e- commerce dealing with fresh food and food items [23–30,32–34] were thoroughly reviewed. Based on the existing literature, service quality attributes related to information quality [23,27], price [23,28,29], product assortment [30], problem resolution [24,25,32], delivery quality [25,32,34], ease of use [24–26], trendiness [35,36], sales promotion [25,28,30,34]), product quality [24,25,33], design [23,27,28] were revealed and related specific measurement items were delineated for item generation.

### 4.1.2. Step 2: Big Data Analysis

The second step in scale development was big data analysis to explore consumers' perceptions of fresh food delivery platforms. Big data can provide a wide range of consumer insights into research areas that conventional research techniques have not identified well [41].

Documents (blog contents) related to fresh food delivery platform services were collected from two major Korean portal websites, "NAVER"(Seongnam, Korea) and "Daum"

(Jeju, Korea), using Textom program—a big data integrated processing solution developed by Loet Leydedorff (Professor at University of Amsterdam) at The IMC Inc. (Daegu, Korea) of Korea. These portal sites are dominant in Korea and more than 70% of Koreans use them when they search and share information online [42]. "Naver" is definitely the largest portal site in South Korea. According to the ICT ministry data, the average daily user number of "Naver" for the last three months of 2021 is 40.3 million, which is about 78% of the total number of Koreans [43]. "Naver" offers various IT services, such as search engines, blog, communities, mail, and more [44]. "Daum" is Korea's second-largest portal sites and it also provides similar services such as "Naver" [45].

The data collection period was from 1 January 2015 to 31 August 2020 and a total of 58,939 documents were gathered. To extract key attributes of fresh food delivery platforms, Term-Frequency-Inverse Document Frequency (TF-IDF) analysis was performed. The TF-IDF analysis is a method that derives keywords based on how frequently they occur in a particular document. The higher this numerical value is, then the more representative the keyword is in a text [46]. Among the extracted words, words related to service quality factors of fresh food delivery platforms (e.g., "fresh food", "shipping", "delivery", "packing", "discount", "taste", etc.) were reflected in the generation of items. Keywords related to delivery service, such as "shipping", "early morning delivery", "delivery, parcel", "today, the day", "arrive", "same-day delivery", "next day", etc.—were extracted. Other keywords—"cold storage", and "ice pack"—were associated with the temperature of the delivery. Keywords which describe products that consumers want or need to purchase through the platforms- "food ingredients", "agricultural products", "vegetable", "frozen food", "meal-kit", etc.- were also extracted. In addition, other keywords worthy of consideration as service quality attributes, such as "discount", "price,", and "trend" were extracted. The text-mining results were considered to be part of the item generation phase. The results of the TF-IDF analysis are displayed in the Appendix A.

### 4.1.3. Step 3: Expert's Interview

The in-depth interviews were conducted with six experts during the period of 1–10 July 2021. In order to conduct this study, we wanted to invite experts who majored in restaurant management and who have worked in the food industry especially in online fresh food commerce for at least 5 years to participate in the interview. For this purpose, among those who graduated from Kyung Hee University or Kyung Hee graduate school, six people who qualified above conditions were contacted and interviewed, and incentives were paid. Two experts are working on an online salad delivery platform, and two experts are in charge of online distribution and sales management of fresh food and food products in the foodservice company. One expert develops refrigerated meal kits and sells them online. The other one is a university lecturer with experience working on a fresh food delivery platform. They have expertise in the fresh food and food products where freshness and temperature control are important. They are well aware of trends in food distribution business and consumer behavior and have ability to provide useful and practical information for this research. Interviewees were asked about service attributes, business experiences, and their opinions of fresh food delivery platform services. For instance, the following questions were asked: "What are the service attributes of fresh food delivery platform services?"; "What do you consider important when selling food products including fresh food online?"; and "What do you think customers care about when buying fresh food online?" In the interview, several significant keywords, such as 'small portion or variety of portion sizes', 'next day delivery', 'delivery damage control', 'temperature-controlled delivery', 'delivery accuracy', 'accurate product information', and 'eco-friendly packaging', were mentioned by experts.

A total of 69 items for measuring service quality were derived based on the results of big data analysis, literature reviews, and in-depth interviews. The initial items were re-examined by six experts to ensure face validity and content validity following the research procedure of Lin and Hsieh (2011) [40] and Lichtenstein et al. (1990) [47]. Six experts who

participated in the expert's interview judged the 69 attributes on a five-point Likert scale anchored by 1 (strongly disagree) and 5 (strongly agree). 14 items with an average score lower than 4 (agree) were removed, per Brakus et al. (2009) [48] study, and a final set of 55 items remained.

*4.2. Study 2: Preliminary Assessment (Scale Refinement and Purification)*

In Study 2, a questionnaire survey was conducted as a preliminary assessment of the 55 item pool generated in Study 1. EFA, reliability analysis, and CFA were performed to refine the items with low reliability validity.

4.2.1. Data Collection and Sampling

To purify and refine the measurement items the survey in Study 2 was conducted. After performing a pre-test with 50 Kyung Hee University students majoring in foodservice management, the main survey was carried out in November 2021. The consumer survey was conducted in South Korea by Macromil Embrain, which is the No.1 online research agency with the largest panel (1.5 million) in South Korea [49]. The questionnaire contained the 55 items from Study 1. Each item was measured using a 7-point Likert scale anchored by 1 (strongly disagree) and 7 (strongly agree). Questionnaires were distributed by an online survey company, to randomly selected respondents who had bought food products through Korean fresh food delivery platform (e.g., Market Kurly, Oasis, Coupang Fresh, and etc.) within the last month. The subjects of this survey are the top 6 down delivery service providers in Korea considering the number of users [50]. These platforms specialize in selling fresh food and food products and provide services that allow consumers receive orders the next day immediately. "Market Kurly" is a pioneer of the fresh food delivery service and has more than 10 million users at the end of 2021 [51,52]. "Coupang Fresh" is the fastest growing online fresh food commerce in South Korea [52].

In the sampling process, in order to determine whether or not they have actually purchased food through a food distribution company, they were asked to select the buying channel they have used for food purchase within the last 1 month. In addition, the respondents who selected online shopping mall among various items such as offline market, TV home shopping channel, and online were extracted. Next, after asking what kind of online shopping mall respondents used to purchase food within 1 month, the respondents who chose to use the online shopping mall as defined in this study were finally selected and participated in the survey. A total of 2079 respondents tried to participate the consumer survey but 1446 unqualified responses were screened out based on screening items that asked which types of online shopping sites respondents had used before. 65 insincere cases were also eliminated.

Accordingly, 550 valid questionnaires were used for data analysis. Around 63.6% of the respondents were female and 36.4% were male. A plurality of respondents were 30–39 years old (39.1%), married (62.7%), and lived in a four person house hold (32.0%). In terms of frequency of purchasing food products on fresh food delivery platforms, 53.7% of respondents made purchases less than once a week and 41.4% made purchases 2–3 times a week. The respondents' demographic profiles are presented in Table 1.

**Table 1.** Profiles of the respondents in Study 2 (*n* = 550).

| Demographics | N | % | Demographics | N | % |
|---|---|---|---|---|---|
| **Gender** | | | **Education** | | |
| Female | 350 | 63.6 | High school or below | 63 | 11.5 |
| Male | 200 | 36.4 | University | 434 | 78.9 |
| **Age** | | | Graduate school or above | 53 | 9.6 |
| 20–29 years old | 83 | 15.1 | **Household size** | | |
| 30–39 years old | 215 | 39.1 | One person (self) | 83 | 15.1 |
| 40–49 years old | 169 | 30.7 | Two persons | 103 | 18.7 |
| Over 50 years old | 83 | 15.1 | Three persons | 156 | 28.4 |

**Table 1.** *Cont.*

| Demographics | N | % | Demographics | N | % |
|---|---|---|---|---|---|
| **Marital status** | | | Four persons | 176 | 32.0 |
| Single | 205 | 37.3 | Five persons | 29 | 5.3 |
| Married | 345 | 62.7 | Six persons or more | 3 | 0.5 |
| **Occupation** | | | **Frequency of purchasing food products through** | | |
| Student | 34 | 4.7 | **the fresh food delivery platform** | | |
| Office worker | 285 | 51.8 | Less than 1 times/week | 296 | 53.8 |
| Self-employed | 32 | 5.8 | 2–3 times/week | 227 | 41.3 |
| Professional | 52 | 9.5 | 4–5 times/week | 22 | 4.0 |
| Housewife | 110 | 20.0 | >6 times/week | 5 | 0.9 |
| Others | 37 | 6.7 | **Primary fresh food delivery service** | | |
| **Family income** | | | Market Kurly | 329 | 59.8 |
| <USD 2000/month | 35 | 6.4 | Coupang Fresh | 172 | 31.3 |
| USD 2000–2999/month | 75 | 13.6 | Oasis | 38 | 6.9 |
| USD 3000–4999/month | 161 | 29.3 | Hello nature | 11 | 2.0 |
| USD 5000–6999/month | 149 | 27.1 | Others (SSG and to home) | 0 | 0 |
| USD 7000–9999/month | 90 | 16.4 | | | |
| ≥USD 10,000/month | 40 | 7.3 | **Total** | **570** | **100** |

#### 4.2.2. Results

Three sets of EFA were obtained using a principal axis factoring method with oblimin rotation. The Kaiser-Meyer-Olkin (KMO) and Bartlett's test of Sphericity (significance $p < 0.05$) were used to evaluate the data's suitability for factor analysis. When KMO ≥ 0.9 is considered very appropriate; $0.8 < \text{KMO} \leq 0.9$ represents fairly appropriate; $0.7 < \text{KMO} \leq 0.8$ represents appropriate; $0.6 < \text{KMO} \leq 0.7$ represents barely appropriate; $0.5 < \text{KMO} \leq 0.6$ is considered inappropriate. The KMO measure of sampling adequacy was 0.953 and Bartlett's test of Sphericity was 16476.095 ($p < 0.001$), indicating the appropriateness of factor analysis. Among the 55 items, 8 items were removed due to having a factor loading below 0.4 in the initial EFA. After two more iterative processes of EFA, 5 items were eliminated due to lower factor loading and 42 items were derived. The factorial structure of the dimensions explained 69.61% of the cumulative variance, which satisfied the desirable criteria of 60% [48] and retained all factors with eigenvalues greater than 1 [53].

Fresh food delivery platform service quality factors with 42 items were used for further testing. Confirmatory factor analysis (CFA) was conducted to improve the psychometric measurement properties of the scale using AMOS. The fit of the measurement model was $\chi^2(791) = 2544.34$, $\chi^2/\text{df} = 3.217$, CFI = 0.891, IFI = 0.891, TLI = 0.881, RMR = 0.081, RMSEA = 0.064. Some fit indices were slightly below the acceptable thresholds. Thus, the standard regression weights and modification indices (MIs) of the measurement items, as well as item context, were inspected. The standardized loading estimate of SQ5-8 was 0.497, less than the suitable criteria of 0.5 [53]. 13 items were deleted based on the MIs. Each item was reviewed to check whether there were other items with overlapping meanings. SQ5-6 and SQ5-5, SQ6-2, SQ6-3, and SQ6-1 had similar question wordings, and three items (SQ5-6, SQ6-2, and SQ6-3) were removed. After a total of 17 items were eliminated in the refinement stage, the measurement fit of the final confirmatory model containing 25 items was more acceptable: $\chi^2(254) = 626.422$, $\chi^2/\text{df} = 2.466$, CFI = 0.958, IFI = 0.958, TLI = 0.950, RMR = 0.059, RMSEA = 0.052. Internal consistency was estimated by Cronbach's alpha, and the values of all domains ranged from 0.718 to 0.892, which is above the threshold of 0.7. In total, 25 items with seven dimensions representing the service quality of fresh food delivery platforms were generated. The final measurement items for Food PlatQual were then determined (Table 2). This study suggested the following labels: "trendiness", "product assortment", "delivery quality", "information quality", "price", "ease of use", and "problem resolution".

**Table 2.** Exploratory factor analysis results for initial measurement items for Food PlatQual (Study 2).

| Items | | Study 2 (*n* = 550) | | |
|---|---|---|---|---|
| | | Mean | SD | EFA Factor Loadings |
| Factor 1: Information quality (α = 0.916) | | | | |
| SQ 1-1 | The platform offers food product storage instruction. | 5.260 | 1.031 | 0.657 |
| *SQ 1-2* | *The platform offers accurate information of food manufacturer.* | *5.316* | *0.998* | *0.656* |
| SQ 1-3 | The product information on the platform is useful. | 5.329 | 0.951 | 0.621 |
| *SQ 1-4* | *The platform offers detailed product information.* | *5.329* | *1.002* | *0.614* |
| SQ 1-5 | The platform offers various products images. | 5.283 | 1.128 | 0.605 |
| SQ 1-6 | The product information on the platform is accurate. | 5.274 | 0.937 | 0.536 |
| SQ 1-7 | The product information on the platform is easy to understand. | 5.400 | 1.008 | 0.496 |
| *SQ 1-8* | *The platform offers product shelf-life information.* | *5.101* | *1.248* | *0.470* |
| Factor 2: Price (α = 0.910) | | | | |
| SQ 2-1 | The price of items sold on the platform is reasonable. | 4.661 | 1.192 | 0.930 |
| *SQ 2-2* | *The platform sells items at a right price.* | *4.707* | *1.187* | *0.929* |
| SQ 2-3 | The price of items sold on the platform is lower than offline-store. | 4.921 | 1.102 | 0.826 |
| SQ 2-4 | The price of items sold on the platform is reasonable considering the quality. | 4.447 | 1.475 | 0.814 |
| Factor 3: Product assortment (α = 0.772) | | | | |
| *SQ 3-1* | *The platform offers several package sizes per each variety of products.* | *5.050* | *1.156* | *0.755* |
| SQ 3-2 | The platform offers a variety of refrigerated and frozen product. | 5.494 | 1.046 | 0.595 |
| SQ 3-3 | The platform offers various vegetable, fruit and grain varieties. | 5.356 | 1.076 | 0.549 |
| SQ 3-4 | The platform offers a variety of imported food products. | 5.118 | 1.202 | 0.500 |
| Factor 4: Problem resolution (α = 0.877) | | | | |
| SQ 4-1 | Consumers can return and exchange product immediately. | 5.327 | 1.082 | 0.916 |
| SQ 4-2 | Consumers can return and exchange product easily. | 5.350 | 1.125 | 0.911 |
| SQ 4-3 | The platform solves customer's problem quickly. | 5.252 | 1.082 | 0.760 |
| *SQ 4-4* | *The platform communicates with customers through social networking services.* | *4.896* | *1.189* | *0.424* |
| Factor 5: Delivery quality (α = 0.899) | | | | |
| SQ 5-1 | The product is kept in a chilled or frozen state during delivery. | 5.603 | 1.135 | 0.870 |
| SQ 5-2 | The platform delivers food at the right temperature. | 5.596 | 1.106 | 0.841 |
| SQ 5-3 | The product is not damaged during delivery. | 5.387 | 1.212 | 0.814 |
| SQ 5-4 | Consumers are satisfied with their delivery system. | 5.570 | 1.120 | 0.790 |
| SQ 5-5 | The platform delivers products accurately. | 5.821 | 0.999 | 0.734 |
| *SQ 5-6* | *Consumers can receive the ordered items on time.* | *5.816* | *1.049* | *0.566* |
| *SQ 5-7* | *The platform uses eco-friendly packaging materials.* | *5.283* | *1.313* | *0.501* |
| *SQ 5-8* | *Consumers can track a status of delivery.* | *5.343* | *1.274* | *0.427* |
| *SQ 5-9* | *The platform delivers items within an adequate period of time.* | *6.096* | *0.946* | *0.418* |
| Factor 6: Ease of use (α = 0.913) | | | | |
| SQ 6-1 | The display pages within the platform are easy to use. | 5.427 | 1.059 | −0.863 |
| *SQ 6-2* | *The layout of the platform is aesthetically appealing.* | *5.250* | *1.105* | *−0.743* |
| *SQ 6-3* | *The display pages within the platform are easy to read.* | *5.458* | *0.995* | *−0.714* |
| SQ 6-4 | Customers can search the items quickly. | 5.432 | 1.058 | −0.678 |
| *SQ 6-5* | *The product image on the platform is aesthetically appealing.* | *5.285* | *1.020* | *−0.586* |
| *SQ 6-6* | *The platform offers information with combination of text and image.* | *5.267* | *1.025* | *−0.555* |
| SQ 6-7 | It is easy to order items on the platform. | 5.889 | 0.860 | −0.497 |

**Table 2.** *Cont.*

| | Items | Study 2 (*n* = 550) | | |
| --- | --- | --- | --- | --- |
| | | **Mean** | **SD** | **EFA Factor Loadings** |
| Factor 7: Trendiness (α = 0.865) | | | | |
| SQ 7-1 | Various new food ingredients are available. | 5.480 | 0.976 | −0.837 |
| SQ 7-2 | Many new products reflecting hot trends are available. | 5.432 | 0.985 | −0.773 |
| SQ 7-3 | Various trending food products are available. | 5.573 | 1.024 | −0.744 |
| SQ 7-4 | *A variety of seasonal food are available.* | *5.645* | *0.870* | *−0.615* |
| Factor 8: Sales promotion (α = 0.829) | | | | |
| SQ 8-1 | The platform manages good incentive program (e.g., coupon and mileage). | 5.363 | 1.174 | 0.652 |
| SQ 8-2 | The platform offers special offers and promotions. | 5.278 | 1.166 | 0.643 |

KMO = 0.953, Bartlett test of Sphericity = 16,476.095 (*p* < 0.001). 42 items derived from three times of EFA. Items in italics are were deleted from the initial pool of 42 items after CFA in Study2; 25 items were remained.

*4.3. Study 3: Scale Validation*

To confirm the 25 Food PlatQual items discovered in Study 2, a second consumer survey was carried out in Study 3. CFA and correlation analysis revealed that the items of the Food PlatQual scale have convergent and discriminant validity.

4.3.1. Data Collection and Sampling

The data for scale validation were collected in December 2021 via an online research company, Macromil Embrain. The 25 items on the questionnaire were measured by a 7-point Likert scale. The second consumer survey targeted a different group of respondents who did not participate in the survey in Study 3 and had experience purchasing food products from Korean fresh food delivery platforms (e.g., Market Kurly, Oasis, Coupang Fresh, and etc.) within the last month. Of the 2379 surveys that were distributed, 1719 cases were screened out and 90 were removed due to incomplete answers. Consequently, 570 responses were used for further data analysis. A majority of respondents were female (67.4%). In terms of age, most respondents were between 30–39 years old (40.0%) or 40–49 years old (30.4%). More than half of the respondents (55.4%) purchased food products via fresh food delivery platforms less than once a week, while 42.8% of them purchased from fresh food delivery platforms 2–3 times a week. The respondents' demographic information is presented in Table 3.

**Table 3.** Profiles of the respondents in Study 3 (*n* = 570).

| Demographics | N | % | Demographics | N | % |
| --- | --- | --- | --- | --- | --- |
| **Gender** | | | **Education** | | |
| Female | 384 | 67.4 | High school or below | 59 | 10.4 |
| Male | 186 | 32.6 | University | 454 | 79.6 |
| **Age** | | | Graduate school or above | 57 | 10.0 |
| 20–29 years old | 84 | 14.7 | **Household size** | | |
| 30–39 years old | 228 | 40.0 | One person (self) | 90 | 15.8 |
| 40–49 years old | 173 | 30.4 | Two persons | 103 | 18.1 |
| Over 50 years old | 85 | 14.9 | Three persons | 193 | 33.9 |
| **Marital status** | | | Four persons | 151 | 26.5 |
| Single | 235 | 41.2 | Five persons | 27 | 4.7 |
| Married | 335 | 58.8 | Six persons or more | 6 | 1.1 |
| **Occupation** | | | **Frequency of purchasing food products through** | | |
| Student | 27 | 4.7 | **the fresh food delivery platform** | | |
| Office worker | 346 | 60.7 | Less than 1 times/week | 316 | 55.4 |
| Self-employed | 24 | 4.2 | 2–3 times/week | 244 | 42.8 |

**Table 3.** *Cont.*

| Demographics | N | % | Demographics | N | % |
|---|---|---|---|---|---|
| Professional | 57 | 10.0 | 4–5 times/week | 9 | 1.6 |
| Housewife | 87 | 15.3 | >6 times/week | 1 | 0.2 |
| Others | 29 | 5.1 | **Primary fresh food delivery service** | | |
| **Family income** | | | Market Kurly | 227 | 39.8 |
| <USD 2000/month | 35 | 6.1 | Coupang Fresh | 285 | 50.0 |
| USD 2000–2999/month | 81 | 14.2 | Oasis | 21 | 3.7 |
| USD 3000–4999/month | 154 | 27.0 | Hello nature | 12 | 2.1 |
| USD 5000–6999/month | 167 | 29.3 | Others (SSG and to home) | 25 | 4.4 |
| USD 7000–9999/month | 101 | 17.7 | | | |
| ≥USD 10,000/month | 32 | 5.6 | **Total** | **570** | **100** |

#### 4.3.2. Data Analysis

The reliability is judged through the Cronbach's alpha value obtained using SPSS version 18. In addition, CFA was conducted to evaluate construct validity, convergent validity, and discriminant validity using AMOS version 18. The sample was divided into two groups and cross-validation was analyzed by performing a Chi-square difference test. Additionally, nomological validity was confirmed through correlation analysis between factors.

#### 4.3.3. Results
#### Reliability and Validity Assessment

The Cronbach's alpha for the seven domains ranged from 0.830 to 0.924, indicating high internal consistency [54]. In order to define construct validity, convergent validity, and discriminant validity, they were examined using CFA. First, the fit of the measurement model was suitable: $\chi^2(254) = 592.319$, $\chi^2/df = 2.332$, CFI = 0.968, GFI = 0.922, IFI = 0.968, TLI = 0.965, RMR = 0.054, RMSEA = 0.048. In addition, standardized factor loading surpassed the acceptable level of 0.50 [53]. Second, convergent validity was estimated by calculating the average variance extracted (AVE) and construct reliability (CR). The AVE values of each construct ranged from 0.580 to 0.815, which exceeded the threshold value of 0.5 [54]. The CR values ranged from 0.805 to 0.929, which is higher than the minimum requirement of 0.7 [53]. Thus, convergent validity was confirmed. Third, discriminant validity was tested by comparing the AVE value and the squared correlation coefficient between a pair of constructs [55]. Since the squared correlation coefficients ranged from 0.122 to 0.568 and did not exceed the AVE values for any construct, discriminant validity was supported. Table 4. presents the results of CFA.

**Table 4.** Reliability and convergent validity of the Food PlatQual (Study 3).

| Items | Study 3 (*n* = 570) | | | | |
|---|---|---|---|---|---|
| | **Mean** | **SD** | **Standardize Factor Loadings** | **AVE** | **CR** |
| Factor 1: Information quality (5 items, α = 0.910) | | | | | |
| SQ 1-1 | 5.094 | 1.034 | 0.805 | | |
| SQ 1-3 | 5.066 | 0.982 | 0.856 | | |
| SQ 1-5 | 5.207 | 1.079 | 0.788 | 0.662 | 0.907 |
| SQ 1-6 | 5.066 | 1.036 | 0.861 | | |
| SQ 1-7 | 5.235 | 0.985 | 0.790 | | |
| Factor2: Price (3 items, α = 0.858) | | | | | |
| SQ 2-1 | 4.642 | 1.148 | 0.822 | | |
| SQ 2-3 | 4.291 | 1.425 | 0.761 | 0.580 | 0.805 |

**Table 4.** *Cont.*

| Items | Study 3 (*n* = 570) | | | | |
|---|---|---|---|---|---|
| | **Mean** | **SD** | **Standardize Factor Loadings** | **AVE** | **CR** |
| SQ 2-4 | 4.694 | 1.111 | 0.891 | | |
| Factor3: Product assortment (3 items, α = 0.849) | | | | | |
| SQ 3-2 | 5.391 | 1.052 | 0.830 | | |
| SQ 3-3 | 5.273 | 1.049 | 0.864 | 0.649 | 0.846 |
| SQ 3-4 | 4.968 | 1.099 | 0.738 | | |
| Factor4: Problem resolution (3 items, α = 0.798) | | | | | |
| SQ 4-1 | | | 0.913 | | |
| SQ 4-2 | 5.210 | 1.180 | 0.901 | 0.815 | 0.929 |
| SQ 4-3 | 5.035 | 1.099 | 0.758 | | |
| Fator5: Delivery quality (5 items, α = 0.911) | | | | | |
| SQ 5-1 | 5.603 | 1.044 | 0.912 | | |
| SQ 5-2 | 5.521 | 1.035 | 0.902 | | |
| SQ 5-3 | 5.377 | 1.164 | 0.765 | 0.678 | 0.913 |
| SQ 5-4 | 5.522 | 1.096 | 0.830 | | |
| SQ 5-5 | 5.714 | 1.052 | 0.812 | | |
| Factor6: Ease of use (3 items, α = 0.924) | | | | | |
| SQ 6-1 | 5.222 | 1.049 | 0.786 | | |
| SQ 6-4 | 5.280 | 1.125 | 0.799 | 0.622 | 0.831 |
| SQ 6-7 | 5.584 | 0.970 | 0.826 | | |
| Factor7: Trendiness (3 items, α = 0.899) | | | | | |
| SQ 7-1 | 5.321 | 0.999 | 0.866 | | |
| SQ 7-2 | 5.278 | 1.032 | 0.849 | 0.739 | 0.894 |
| SQ 7-3 | 5.447 | 1.049 | 0.881 | | |

$\chi^2(254) = 592.319$, $\chi^2/\mathrm{df} = 2.332$, CFI = 0.968, GFI = 0.922, IFI = 0.968, TLI = 0.965, RMR = 0.054, RMSEA = 0.048.

Cross Validity

To ensure the validity of the measurement items, two invariance tests were performed across age groups and randomly selected samples. Cross validity was verified by dividing the research sample into two groups and checking the Chi-square difference of the measurement models between the two groups. First, the respondents were split into two groups by age: a younger group of respondents in their 20s-30s (*n* = 312) and an older group over 40 (*n* = 258). The results showed that the measurement model did not differ between the two groups based on respondents' age ($\Delta\chi^2(\Delta\mathrm{df} = 17) = 21.000$; $p = 0.226$). In addition, the sample was randomly split into two groups (group A, *n* = 285; group B, *n* = 285) and no significant difference was observed between the two groups ($\Delta\chi^2(\Delta\mathrm{df} = 17) = 22.871$; $p = 0.153$). Therefore, the measurement model was invariant for different groups and confirmed the cross validity of the 7 dimensional structure of Food PlatQual (see Table 5.).

**Table 5.** Results of Cross Validations.

| Goodness-Fit Indices | Measurement Models | | | |
|---|---|---|---|---|
| | **Age (20s–30s = 312; Over 40s = 258)** | | **Random Split (Group A = 285; Group B = 285)** | |
| | **Unconstrained** | **Measurement Model** | **Unconstrained** | **Measurement Model** |
| RMSEA | 0.039 | 0.039 | 0.038 | 0.038 |
| RMR | 0.057 | 0.062 | 0.057 | 0.060 |
| CFI | 0.958 | 0.958 | 0.960 | 0.960 |

**Table 5.** *Cont.*

| Goodness-Fit Indices | Measurement Models | | | |
|---|---|---|---|---|
| | Age (20s–30s = 312; Over 40s = 258) | | Random Split (Group A = 285; Group B = 285) | |
| | Unconstrained | Measurement Model | Unconstrained | Measurement Model |
| GFI | 0.883 | 0.880 | 0.883 | 0.881 |
| IFI | 0.958 | 0.958 | 0.961 | 0.960 |
| TLI | 0.951 | 0.952 | 0.953 | 0.954 |
| $\chi^2$ | 963.209 | 984.209 | 933.329 | 956.200 |
| Df | 512 | 529 | 512 | 529 |
| $\chi^2/df$ | 1.881 | 1.861 | 1.823 | 1.808 |

Nomological Validity

Nomological validity was examined by testing the correlations between the 7 dimensions of Food PlatQual and theoretically related variables. Much of the previous literature has proven that attitude, satisfaction, and behavioral intentions are affected by service quality [56–59]. To assess the nomological validity of Food PlatQual, correlations between the 7 dimensions and attitude, satisfaction, and behavioral intentions were analyzed. Three related variables were derived from previous research [60–63] and all of them were measured with four items using a 7-point Likert scale. As shown in Table 6, all variables were positively and significantly correlated ($p < 0.001$). Therefore, the nomological validity of Food PlatQual was confirmed.

**Table 6.** Correlations of seven dimensions of Food PlatQual with attitude, satisfaction, and behavioral intention.

| | INQ | PRI | PRA | PRE | DEQ | EOU | TRN |
|---|---|---|---|---|---|---|---|
| **SAT** | 0.635 ** | 0.470 ** | 0.523 ** | 0.575 ** | 0.540 ** | 0.651 ** | 0.562 ** |
| **ATT** | 0.649 ** | 0.415 ** | 0.505 ** | 0.550 ** | 0.548 ** | 0.669 ** | 0.554 ** |
| **INT** | 0.614 ** | 0.438 ** | 0.508 ** | 0.576 ** | 0.535 ** | 0.646 ** | 0.541 ** |

** $p < 0.01$. INQ—information quality; PRI—price; PRA—product assortment; PRE—problem resolution; DEQ—delivery quality; EOU—ease of use; TRN—Trendiness; SAT—satisfaction; ATT—attitude; INT—behavioral intention.

## 5. Discussion and Implications

The major purpose of this study was to develop and validate a fresh food delivery platform service quality scale, called Food PlatQual. Several significant findings will be discussed below. First, the Food PlatQual scale consisting of 7 dimensions was developed through a systematic procedure: item generation via big data analysis and expert interviews, preliminary assessment based on consumer survey data, and scale validation based on data from a second consumer survey. As a result, a scale with verified reliability, convergent validity, discriminant validity, and nomological validity was developed consisting of "information quality", "price", "product assortment", "problem resolution", "delivery quality", "ease of use", and "trendiness".

Second, "information quality" emerged as a factor of service quality. This result is in accordance with studies by Lee (2018) [25] and Robina-Ramírez et al. (2020) [26]. Information quality is the extent to which a fresh food platform provides accurate, understandable, and useful information using various media formats (e.g., image or video). In particular, it is important to provide high-quality information to reduce consumer anxiety because the quality of fresh ingredients such as vegetables, meat, and fish are not consistent such as manufactured goods.

Third, "price" appeared as a dimension of Food PlatQual. "Price" refers to how reasonable the prices of products sold by fresh food platforms are. Kim and Kim (2019) [24] stated that high prices make consumers hesitant to purchase food online. Lee (2018) [25], Singh (2019) [31], and Ji et al. (2019) [33] also noted the importance of price. However, this

dimension showed relatively lower mean values than other dimensions. This is presumably due to the fact that fresh food platforms mainly deal with premium and organic foods, and consumers do not expect the same level of financial benefits compared to offline stores.

Fourth, "product assortment", another service quality factor focused on the product for sale, refers to providing consumers with a choice of different types and sizes of food products. As Park et al. (2019) [32] observed, consumers use these platforms to purchase products that are difficult to buy offline. Consumers are aware that they could purchase any number of various processed foods and agricultural products. Singh (2019) [31] also pointed out that product assortment is an important factor affecting intentions to use an online grocery platform. Given the results of the current study and previous literature, it appears that consumers expect to easily purchase imported food products and foreign ingredients from fresh food platforms.

Fifth, "problem resolution" is an important factor in online commerce, including fresh food delivery platforms. "Problem resolution" refers to the quality of a service's ability to easily and immediately handle consumer problems such as returns and refunds. This dimension was mentioned in previous research as well [24,31,33]. Singh (2019) [31] argued that easy return, refund, and cancellation policies are crucial task-related expectations and a significant indicator of service excellence. In Kim and Kim's (2019) [24] study, consumers cited difficult refund and exchange procedures as a disadvantage of purchasing food online. No-hassle return is the key to taking fresh food delivery platform businesses to the next level.

Sixth, "delivery quality" had the highest mean score and was definitely a crucial factor in assessing service quality. Delivery quality refers to whether a service delivers products accurately as ordered, at the proper temperature, and without being damaged. The results are in line with previous literature [24,25] that emphasized the importance of the delivery aspects of online food purchases. In other words, products should be delivered to the right person, at the right place, at the right time, and at the right temperature to ensure the safety and freshness of the products.

Seventh, "ease of use" is the degree to which consumers can easily browse, order, and pay for products. As expected, "ease of use" was included as a component of fresh food platform service quality, similar to other online environments in prior studies [31,33]. The result suggests that ease of use is the most basic factor that must be satisfied for technology usage and consumer satisfaction. These results reconfirmed that when consumers evaluate service quality, service performance is not the only factor to be taken into account; a smooth service delivery process is also an important consideration.

Lastly, "trendiness", which indicates the availability of new and trending food products and ingredients, also had a higher mean value. This study found that one of the attributes that consumers expect from fresh food platforms is access to trending products and the newest food items. This result implies that if a fresh food platform sells trendy products or ingredients, then consumers perceive the platform as playing a role in leading food trends. Although trendiness had been mentioned in few studies [23], it has not been largely addressed. The results of this study make it clear that trending products are an important variable that need to be considered in future research.

### 5.1. Theoretical Implications

This study contributes to the existing literature on online food purchasing in several ways. First, this study has crucial theoretical implications in that it followed a systematic process and developed a sound instrument to measure the service quality of online fresh food delivery platforms. As the online fresh food market rapidly grows due to the development of cold chain technology and the outbreak of COVID-19, many researchers have explored online food buying behaviors. However, a service quality scale for online fresh food delivery services had not been developed before this study. Therefore, Food PlatQual could be used for further research related to the service quality of fresh food delivery services.

Second, this study contributes methodologically to existing scale development research by combining big data analysis and conventional scale development procedures. Big data could supplement interview and survey methods that supply more limited information and provide valuable insight about consumers' perceptions and desired service attributes. The methodology of the current study, which grafted more traditional and contemporary analysis methods, contributes to the advancement of scale development methodology.

Lastly, this study makes a theoretical contribution by discovering that "trendiness", which had not been widely examined in previous research, is a new service quality component. It can be inferred that this result reflects the lifestyle of contemporary consumers who value novelty, fashion, and trends since the development of social networking services. "Trendiness" needs to be considered not only in the field of fresh food delivery platform businesses but in other areas as well when observing consumer behaviors.

### 5.2. Managerial Implications

This study offers several managerial implications. First, Food PlatQual could be used as a guideline for online food retail entrepreneurs to satisfy existing consumers and attract potential consumers. E-commerce in the fresh food sector is growing rapidly and consumer buying patterns are changing from offline to online. In addition, the needs of consumers are changing rapidly. Therefore, understanding what consumers want and developing services that can satisfy their needs is the way to survive in a competitive marketplace.

Second, understanding the service quality of fresh food delivery platforms will be helpful to restaurant entrepreneurs interested in developing meal kits to create new revenue. Restaurants will be able to utilize the fresh food delivery platform as a product distribution channel, or some of them will plan to develop their own platform. Therefore, research on the fresh food delivery platform can contribute to the selection or development of distribution networks for the foodservice companies.

Third, it is an interesting result that consumers expect to be able to purchase trending products from fresh food delivery platforms. This could be a distinctive feature that distinguishes fresh food platforms from general online grocery stores. Hence, in order to obtain a competitive advantage, it is necessary to continuously fortify an innovative and trendy image and provide various opportunities for consumers to experience the latest food trends.

Fourth, as predicted, delivery quality is an important component of online commerce, especially when buying fresh food. There are three main issues when it comes to delivery service: temperature control, damaged packages, and delivery accuracy. Given the character of fresh food, temperature-controlled delivery is a critical factor that affects a product's freshness and safety. Companies must implement elaborately designed delivery service strategies and continue to invest in temperature-controlled packaging technology. Further, it is necessary to find ways to reduce the frequency of handling to avoid a higher chance of damage to the delivery items and to load cargo properly with the help of shipping experts. In addition, providing consumers with real-time tracking information from departure to arrival would improve the accuracy of deliveries.

Lastly, platforms should provide enriching information that combines different types of content, such as text, audio, images, or video, and interactively communicate with consumers. Providing such information is critical to attract potential consumers who are concerned about product quality and overcome the fact that the quality of fresh food is not consistent. Additionally, active use of multimedia information will help fresh food delivery platforms present a more innovative image.

### 5.3. Limitations and Future Research

Despite its theoretical and managerial implications, this study has a few limitations as well. First, this study collected data solely from South Korea. Therefore, it is necessary to gather data from other countries and compare the results to verify the generalizability of the scale. Second, this study thoroughly identified service quality attributes based

on qualitative and quantitative methods, but items related to product quality, such as, freshness of food, taste, and safety were refined in the analysis process and not included in Food PlatQual. Since the freshness and safety of food are closely related to delivery quality such as delivery without a damage, delivery temperature, and delivery time, it is presumed that some of them were reflected in the delivery quality dimension. Moreover, it is thought that this is because consumers recognize that delivery quality is more important than product quality when using fresh food delivery platforms. This point might imply that consumer needs when using an offline store and when using an online store may be different. In addition, the fresh food delivery platforms are playing the role of a retailer rather than a producer of fresh food or food products on a food distribution platform. Therefore, it is assumed that the items of product quality had refined from the service quality items that the fresh food delivery platform should have. However, since the product quality is still an important part in overall e-commerce, it is necessary to re-examine these items in future studies.

Lastly, the form of various fresh food delivery platforms, including the product categories and consumer services offered differ across world. Further, these services are constantly evolving and diversifying. Therefore, it is necessary to examine the differences between various types of platforms in future research.

## 6. Conclusions

This study developed the scale for measuring the service quality of fresh food delivery platforms–Food PlatQual. Food PlatQual consisting of 7 dimensions of 25 items were developed through big data analysis and the conventional scale development process. The results from our study suggested the following labels for seven dimensions: "information quality", "price", "product assortment", "problem resolution", "delivery quality", "ease of use", and "trendiness". This is the first study to systematically develop the service quality scale for the fresh food delivery platform, and it provides various theoretical implications for future research as well as managerial implications for practitioners.

**Author Contributions:** J.-W.K. has worked on the conceptual development of the manuscript and data collection and analysis and also wrote the manuscript. Y.N. has reviewed, edited, and offered overall guidance for publishing the manuscript. All authors have read and agreed to the published version of the manuscript.

**Funding:** This work was supported by the Ministry of Education of the Republic of Korea and the National Research Foundation of Korea (NRF-2020S1A5B5A16083048).

**Institutional Review Board Statement:** Not applicable.

**Informed Consent Statement:** Not applicable.

**Data Availability Statement:** Not applicable.

**Conflicts of Interest:** The authors declare no conflict of interest.

## Appendix A  The Result of TF-IDF Analysis

| Word | TF-IDF | Word | TF-IDF | Word | TF-IDF | Word | TF-IDF | Word | TF-IDF |
|---|---|---|---|---|---|---|---|---|---|
| fresh food | 20,867.98 | home | 8869.54 | packing | 6489.81 | meat | 5290.37 | condition | 4654.48 |
| shipping | 16,572.27 | offline | 8210.13 | grocery shopping | 6460.71 | taste | 5226.15 | food ingredients | 4604.83 |
| product | 16,400.79 | supermarket | 7903.20 | distribution | 6435.47 | mobile | 5205.15 | sales | 4583.47 |

| Word | TF-IDF | Word | TF-IDF | Word | TF-IDF | Word | TF-IDF | Word | TF-IDF | Word | TF-IDF |
|---|---|---|---|---|---|---|---|---|---|---|---|
| purchase | 15,891.74 | vegetable | 7796.07 | launch | 6313.91 | growth | 5083.27 | shopping mall | 4483.05 | | |
| online | 13,400.63 | increase | 7427.99 | side dishes | 6137.42 | frozen food | 5052.81 | morning | 4413.53 | | |
| Market Curly | 12,706.43 | Covid19 | 7404.27 | Naver | 6106.55 | expansion | 4930.13 | delivery boundaries | 4389.95 | | |
| Coupang | 12,558.47 | fruit | 7165.54 | food | 6015.74 | supply | 4924.92 | safety | 4360.99 | | |
| early morning delivery | 11,191.11 | E-MART | 6969.87 | discount | 5966.02 | meal-kit | 4867.23 | major su-permarket | 4347.92 | | |
| service | 11,024.56 | market | 6883.71 | today | 5823.19 | same-day delivery | 4857.12 | Internet | 4164.02 | | |
| customer | 10,479.86 | recommend | 6727.06 | ice pack | 5565.97 | trend | 4855.89 | online grocery shopping | 4035.93 | | |
| delivery | 10,238.61 | price | 6878.32 | food product | 5522.11 | time | 4835.33 | agricultural products | 4000.94 | | |
| order | 9955.87 | shopping | 6721.71 | the day | 5503.01 | daily necessity | 4815.89 | cold storage | 3969.49 | | |
| selling | 9710.02 | arrive | 6717.82 | 2020 | 5478.06 | necessary | 4810.42 | cook | 3884.64 | | |
| parcel service | 9020.28 | use | 6665.44 | processed food | 5420.44 | consumption | 4770.03 | convenience | 3875.27 | | |
| utilize | 8952.72 | Rocket Fresh | 6510.38 | ingredients | 5312.96 | Untact | 4655.02 | early morning | 3846.81 | | |

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
