# Peer review of "Measuring the Service Quality of Fresh Food Delivery Platforms: Development and Validation of the “Food PlatQual” Scale"

_sustainability, doi:10.3390/su14105940_

Round 1
Reviewer 1 Report
Please refer to the attached file

Author Response
Thanks for taking your valuable time to review this article.
The authors have tried to improve the quality of the paper based on your advice.
- Living with the covid 19 has become a new norm. Since there is no more movement restriction order (MCO). Thus, some information in the introduction and background section should be updated on this matter. It has been more than two years since the covid 19 affected consumers worldwide.
â–¶Thanks for your comment.
Following your advice, the authors revised the article to emphasize that the business will continue after COVID-19. Author updated the introduction part as follows on page 1.
(on page 1)
Due to time savings and convenience, several experts predict that the growth of the fresh food delivery platform business will continue even after pandemic is over [4, 5, 6]. Amazon’s global sales in the food category are predicted to increase to $26.7 billion by 2026 from $14.5 billion in 2021. Amazon has been investing in its online grocery business, and the Amazon Fresh service has been expanding in international markets as well [4]. Further, according to the report by McKinsey & Company (2021) [5], approximately 70% of respondents’ preferences for fresh food shopping online changed after the COVID 19 outbreak.
- Before the pandemic of Covid 19, online groceries or fresh food shopping platforms and deliveries systems were available in many countries. Different countries may have varying levels of service quality when it comes to online shopping platforms and food delivery. This will be determined by how long the platforms have been in operation, the size of the business, and what is available to customers. Thus, some background information on where the study took place is crucial for readers. Please clarify.
â–¶Thanks for pointing this issue out.
The authors stated the background information in the introduction part as follows on page 1.
(on page 1)
This study targets South Korea, an advanced IT country with an internet usage rate of approximately 92%[7]. In Korea, online shopping has already become a daily life, and the online fresh food purchase rate is also high. Online shopping for fresh food market in Korea stood at $2.02 billion in 2020 and is forecast to grow to $10 billion in 2023 [8]. Therefore, various and practical implications can be obtained by conducting research on fresh food delivery platforms in Korea.
- Are the online delivery platforms in this study is referring to standalone delivery services or online fresh food retailers that provide the delivery service? The word ‘platform’ is quite misleading. It is unclear, and please explain. It is not in line with the data collected in the methodology part.
â–¶In this study, the fresh food delivery platform is defined as an online fresh food retailers that provide the delivery service. The fresh food delivery platform is an e-commerce specializing in the fresh food category [16]. This platform sells food products, fresh foods (e.g., vegetables, fruits, meat, and fish), refrigerated and frozen foods, and other groceries, and offers next-day door to door delivery service [8].
According to Abdul Halim et al. (2020), platform is an overall systems and interfaces that form a commercial network or market facilitating business-to-customer (B2B), business-to-customer (B2C) or even customer-to-customer (C2C) transactions. The subjects of this study such as Market Kurly, Coupang Fresh, and Oasis were stated as fresh food delivery platform in news and papers. Therefore, this study adopted this term.
references :
Korean Herald. http://www.koreaherald.com/view.php?ud=20210711000179
Pulse News Korean. https://pulsenews.co.kr/view.php?year=2021&no=710014
Almunawar, M. Ali, M., Lim, S.(2021). Handbook of Research on Innovation and Development of E-Commerce and E-Business in ASEAN (2 Volumes)
- Most online supermarkets or retailers have their own delivery services, and the shopper doesn’t have a choice in selecting the delivery services. It is all bundled together. Thus, this study refers to the delivery services or fresh food stores in the area with delivery door to door? Please justify.
Line 883
A fresh food delivery platform is an online service that delivers fresh foods, such as vegetables, fruits, meat, fish, processed foods, and other groceries, to a consumer’s doorstep.
Line 131
Fresh food delivery platforms offer not only products that consumer needs but also information.
As an online fresh food shopper - I am selecting the online fresh food store based on the product assortments, not the delivery platform. Because the food delivery method is predetermined by the store when we choose to order the fresh food, please correct me if I’m wrong.
â–¶Thanks for your valuable comments. The fresh food delivery platform is a service that sells fresh and various foods and delivers them to consumers the next day. These platforms are not local retailers, but they deliver products through a logistics system to consumers across the country.
The meaning of the word "delivery" in the "fresh food delivery platform" implies ‘providing a delivery
service’, not selecting or purchasing a delivery service. Innovative services that deliver fresh food to right in front of consumer’s home the next day based on advanced distribution technology away from past services that delivered only products that did not require temperature control is the crucial advantage of these platforms. So these services are usually called "fresh food delivery platforms."
To supplement this perspective, the authors gave more specific definition of the fresh food delivery platform in the main document as below on page 2.
(on page 2)
A fresh food delivery platform is an e-commerce specializing in the fresh food category [16]. This platform sells food products, fresh foods (e.g., vegetables, fruits, meat, and fish), refrigerated and frozen foods, and other groceries, and offers next-day door to door delivery service [8].
- The author has already chosen the primary fresh food delivery service available in the study area in the methodology section. However, whether these platforms are online retail stores with delivery services or standalone fresh food delivery services is still unclear.
- Market Kurly
- Coupang Fresh
- Oasis
- Hello nature
- Others
â–¶Thanks for your comment. For better understanding of the research, the authors revised the definition of the fresh food delivery platform in the literature background section as above (your comments 3~4) and added information about these Korean fresh food delivery platforms in the data collection and sampling section as belows on page 7.
(on page 7)
These platforms specialize in selling fresh food and food products and provide services that allow consumers receive orders the next day immediately. “Market Kurly” is a pioneer of the fresh food delivery service and has more than 10 million users at the end of 2021 [51, 52]. “Coupang Fresh” is the fastest growing online fresh food commerce in South Korea [52].
- It is critical to provide background information about these companies, the area where all the services are available to the respondents, and which region the study has been conducted. The readers for this journal are from worldwide and might not be familiar with listed service providers.
â–¶We really appreciated your valuable comment.
The authors described about these services in more detail for unfamiliar readers in the theoretical background part (page 1) and data collection and sampling section (page6~7).
(on page1)
This study targets South Korea, an advanced IT country with an internet usage rate of approximately 92%[7]. In Korea, online shopping has already become a daily life, and the online fresh food purchase rate is also high. Online shopping for fresh food market in Korea stood at $2.02 billion in 2020 and is forecast to grow to $10 billion in 2023 [8]. Therefore, various and practical implications can be obtained by conducting research on fresh food delivery platforms in Korea.
(on page 6)
Documents (blog contents) related to fresh food delivery platform services were collected from two major Korean portal websites, “NAVER” and “Daum,” using Textom─a big data integrated processing solution developed by Loet Leydedorff (Professor at University of Amsterdam) at The IMC Inc. of Korea. These portal sites are dominant in Korea and more than 70% of Koreans use them when they search and share information online [42].
(on page 7)
After performing a pre-test with 50 Kyung Hee University students majoring in food-service management, the main survey was carried out in November, 2021. The consumer survey was conducted in South Korea by Macromil Embrain which is the No.1 online research agency with the largest panel (1.5 million) in South Korea [49]. The questionnaire contained the 55 items from Study 1. Each item was measured using a 7-point Likert scale anchored by 1 (strongly disagree) and 7 (strongly agree). Questionnaires were distributed by an online survey company, to randomly selected respondents who had bought food products through Korean fresh food delivery platform (e.g., Market Kurly, Oasis, Coupang Fresh, and etc.) within the last month. The subjects of this survey are the top 6 down delivery service providers in Korea considering the number of users [50]. These platforms specialize in selling fresh food and food products and provide services that allow consumers receive orders the next day immediately. “Market Kurly” is a pioneer of the fresh food delivery service and had more than 10 million users at the end of 2021 [51, 52]. “Coupang Fresh” is fastest growing online fresh food commerce in South Korea [52].
- Are there any differences between standard food delivery and fresh food delivery?
â–¶There is a difference between regular food delivery and fresh food delivery. In the case of ordinary foods, the effect of temperature on freshness is less, but in the case of fresh foods, if the temperature is not maintained well, freshness or quality decreases. Fresh foods require more attention to packaging, delivery methods, delivery temperature, and delivery time, etc.
Methodology
- It is unclear where the study has been conducted. Korea? If so, I suggest some background information in the introduction, not just in the methodology.
â–¶Thanks for your sincere comment. This study was conducted in South Korea. To clear the background of this research, the country in which the study was conducted was described in the ‘Introduction’ on page 1.
(on page 1)
This study targets South Korea, an advanced IT country with an internet usage rate of approximately 92%[7]. In Korea, online shopping has already become a daily life, and the online fresh food purchase rate is also high. Online shopping for fresh food market in Korea stood at $2.02 billion in 2020 and is forecast to grow to $10 billion in 2023 [8]. Therefore, various and practical implications can be obtained by conducting research on fresh food delivery platforms Korea.
Line 217
the IMC of Korea - meaning?
â–¶ IMC is the name of company. The authors revised the part describe about textom as below on page 6.
(on page 6)
Documents (blog contents) related to fresh food delivery platform services were collected from two major Korean portal websites, “NAVER” and “Daum,” using Textom─a big data integrated processing solution developed by Loet Leydedorff (Professor at University of Amsterdam) at The IMC Inc. of Korea.
Line 67
Using big data, this study intended to reveal how consumers perceive fresh food delivery platforms in their daily lives and then reflect these results in the item generation phase.
- The two websites’ details. Please give more descriptions of the websites, even though the name cannot be revealed.
Line 214
Documents related to fresh food delivery platform services were collected from two major Korean portal websites, “NAVER” and “Daum,” using Textom─a big data integrated processing solution developed by Loet Leydedorff (Professor at University of Amsterdam) at the IMC of Korea
â–¶Thanks for your advice. The authors provided more descriptions of these websites in the main document as below.
(on page 6)
Documents (blog contents) related to fresh food delivery platform services were collected from two major Korean portal websites, “NAVER” and “Daum,” using Textom─a big data integrated processing solution developed by Loet Leydedorff (Professor at University of Amsterdam) at The IMC Inc. of Korea. These portal sites are dominant in Korea and more than 70% of Koreans use them when they search and share information online [42]
- Food PlatQual – simplified terms? Is it based on? Justification needed.
Line 154
Thus, this research attempted to fill this gap by identifying service attributes and developing a fresh food 185 delivery platform service quality scale─Food PlatQual.
â–¶Just as the researcher made names to the scales (SERVQUAL, ES-QUAL, SITEQUAL, SSTQUAL) developed in the previous study, in this study, the scale measuring the service quality of the fresh food delivery platform was named in order to be concise.
- Step 2: Experts interview.
Line 237
To generate items for Food PlatQual, relevant previous studies on online grocery shopping [16, 19, 20, 24, 25, 26] were reviewed.
Line 242 -245
Interviewees were asked about service attributes, business experiences, and their opinions of fresh food delivery platform services. For instance, the following questions were asked: “What are the service attributes of fresh food delivery platform services?”; “What do you consider important when selling food products online?”; and “What do you think customers care about when buying food online?”
Contrary to the title, it appears to be only about food delivery platforms when it also asked food they sell in the interview questions.
â–¶Thanks for your useful comment.
The authors had asked six experts a wide range of questions, from buying food to buying fresh food in the experts’ interview. Like your advice, author also thinks it will confuse the reader by not clearly stating examples of interview questions. Therefore, author revised the questions as below.
(on page 6)
For instance, the following questions were asked: “What are the service attributes of fresh food delivery platform services?”; “What do you consider important when selling food products including fresh food online?”; and “What do you think customers care about when buying fresh food online?”
- There was no description of how the 62 items were derived from the interviews. What were the data analysis methods used to obtain it? How were the 62 items reduced to 55 items?
Line 250-252
Six experts judged the 69 attributes on a five-point Likert scale anchored by 1 (strongly disagree) and 5 (strongly agree). Items with an average score lower that 4 (agree) were removed, per Brakus et al.’s (2009) [36] study, and a final set of 55 items remained
- Who and how many were the respondents? How were the average mean scores obtained? Justify
â–¶Thanks for your comments 12 and 13.
As authors explained in the main document, six experts judged the 69 attributes, derived based on the results of big data analysis, literature reviews, and in-depth interviews, on a five-point Likert scale anchored by 1 (strongly disagree) and 5 (strongly agree). The survey was conducted briefly by six experts to review the survey items prior to the main survey. With reference to Brakus et al.’s (2009) methodoloy, items with an average score of 4 or less evaluated by 6 experts were deleted. Thus, 14 items with an average score lower than 4 (agree) were removed and a final set of 55 items remained.
For a better understanding, authors revised the sentence as follows.
(on page 7)
Six experts judged the 69 attributes on a five-point Likert scale anchored by 1 (strongly disagree) and 5
(strongly agree). 14 items with an average score lower than 4 (agree) were removed, per Brakus et al.’s
(2009) [45] study, and a final set of 55 items remained.
- Study 2
50 students majoring in foodservice management – who they are? In Korea?
The authors added more information about 50 students participated in the pre-test in the document as below on page 7.
(on page 7)
After performing a pre-test with 50 Kyung Hee University students majoring in food-service management, the main survey was carried out in November, 2021.
- A fresh food delivery platform (e.g., Market Kurly, Oasis, Coupang Fresh, and etc.). Where are the listed platforms located? How were the platforms selected? Are they only providing delivery services or also selling the foods?
â–¶Thanks for your comment. These service platforms are located in Korea. On page 10, the authors mentioned that these platforms are the top 7 companies in Korea This information was provided in the second consumer research section (p. 10), but the authors have moved this phrase to the first consumer survey section (p.7), which is presented first.
These services sell fresh food and other food category things and delivered ordered items to customers. We provided more specific information about the subjects of this study on page 7.
(on page 7)
The subjects of this survey are the top 6 down delivery service providers in Korea considering the number of users [50]. These platforms specialize in selling fresh food and food products and provide services that allow consumers receive orders the next day immediately. “Market Kurly” is a pioneer of the fresh food delivery service and has more than 10 million users at the end of 2021 [51, 52]. “Coupang Fresh” is the fastest growing online fresh food commerce in South Korea [52].
Line 265- 270
Questionnaires were distributed by an online survey company to randomly selected respondents who had bought food products through a fresh food delivery platform (e.g., Market Kurly, Oasis, Coupang Fresh, and etc.) within the last month. A total of 2,079 respondents accessed the questionnaire but 1,446 unqualified responses were screened out based on screening items that asked which types of online shopping sites respondents had used before. 65 uncompleted questionnaires were also eliminated.
- The sampling methods and the procedures were not clear in the methodology. Please clarify and justify.
â–¶Thanks for your comment. To clarify methodology, authors gave more detailed information about online research company and sampling process in the main document as follow on page 7.
(on page 7)
The consumer survey was conducted in South Korea by Macromil Embrain which is the No.1 online
research agency with the largest panel (1.5 million) in South Korea [49]. The questionnaire contained
the 55 items from Study 1. Each item was measured using a 7-point Likert scale anchored by 1
(strongly disagree) and 7 (strongly agree). Questionnaires were distributed by an online survey
company, to randomly selected respondents who had bought food products through Korean fresh food
delivery platform (e.g., Market Kurly, Oasis, Coupang Fresh, and etc.) within the last month. The
subjects of this survey are the top 6 down delivery service providers in Korea considering the number
of users [50]. These platforms specialize in selling fresh food and food products and provide services
that allow consumers receive orders the next day immediately. “Market Kurly” is a pioneer of the fresh
food delivery service and had more than 10 million users at the end of 2021 [51, 52]. And “Coupang
Fresh” is fastest growing online fresh food commerce in South Korea [52].
In the sampling process, in order to determine whether or not they have actually purchased food
through a food distribution company, they were asked to select the buying channel they have used for
food purchase within the last 1 month. And the re-spondents who selected online shopping mall among
various items such as offline market, TV home shopping channel, and online were extracted. Next,
after asking what kind of online shopping mall respondents used to purchase food within 1 month, the
respondents who chose to use the online shopping mall as defined in this study were finally selected
and participated in the survey. A total of 2,079 respondents to participate the consumer survey but
1,446 unqualified responses were screened out based on screening items that asked which types of
online shopping sites respondents had used before. 65 insincere cases were also eliminated.
- Please separate the discussion and the conclusion or provide a separate heading for the conclusion.
â–¶Thanks for your suggestion. The authors separated the discussion and conclusion. Please see page 17.
(on page 17)
- Conclusion
This study developed the scale for measuring the service quality of fresh food de-livery platforms─Food PlatQual. Food PlatQual consisting of 7 dimensions of 25 items were developed through big data analysis and the conventional scale development process. The results from our study suggested the following labels for seven dimensions: “information quality,” “price,” “product assortment,” “problem resolution,” “delivery quality,” “ease of use,” and “trendiness.” This is the first study to systematically develop the service quality scale for the fresh food delivery platform, and it provides various theoretical implications for future research as well as managerial implications for practitioners.
- It is necessary to make minor corrections to English to improve the flow.
â–¶ The authors checked grammatical errors and will ask a native English speaker to proofread the paper.
Thank you very much for the reviewer’s valuable comments for the improvement of this manuscript.

Reviewer 2 Report
First of all, I congratulate the authors on the material they have produced. I think that the topic they have started to address is interesting and will become more and more important regardless of the COVID pandemic, so that they have been able to create a very useful consumer-oriented scale for future research.
My problem with the article they have produced is in the area of inlogical construction. As I read the paper, the thought arc, the logical structure did not come through, which made it difficult to understand the article, so I would suggest that it should be rethought in this respect.
The logical structure of the introduction chapter does not really come through either. At first, the presentation of the market trends and the situation caused by the COVID epidemic and thus the penetration of the online market is understandable, but the content between lines 39 and 44 is for me disjointed and not comprehensible. In lines 45-46 the authors write: 'there has been a great deal of research related to service quality in the context of online food purchases' - not a single one is cited. In the literature review, I felt there were several areas where references were lacking, and I would rework this chapter in this respect
Material and Methods chapter:
I do not consider the description of study 1 to be sufficiently thorough. The description of the Big Data analysis is acceptable, but the expert interview and the literature review are particularly weak. In the description of the expert interview, they start by analysing literature and then cite 6 articles. This gives the impression that 6 articles were considered as part of the literature review when constructing the scale. But they also did interviews on the side. This is not good. Besides, very little is revealed about the interviews.
The results chapter should be consolidated in terms of the presentation of indicators. Some indicators are simply described as such (KMO and Bartlett's test), while others are nicely explained as to whether a given value is appropriate or not.
It was confusing to me that the material and methods chapter does not yet focus on the material and methods issues of the third study, but does within the results, and then there is a results chapter within the results. I suggest that what is material and method should be in that chapter and what is results should be in the results chapter.
However, the strength of the article is the discussion and further research chapters.

Author Response
Thanks for taking your valuable time to review this article.
The authors have tried to improve the quality of the paper based on your advice.
First of all, I congratulate the authors on the material they have produced. I think that the topic they have started to address is interesting and will become more and more important regardless of the COVID pandemic, so that they have been able to create a very useful consumer-oriented scale for future research.
My problem with the article they have produced is in the area of inlogical construction. As I read the paper, the thought arc, the logical structure did not come through, which made it difficult to understand the article, so I would suggest that it should be rethought in this respect.
â–¶ Thanks for your advice. This study consisted of three studies. Therefore, the flow of article could be somewhat different with other articles. As your comment, we tried to revise the construct (add 4.1.1. before 4.1.2. big data analysis). We constructed this by referring to the previous literatures (Kim et al., 2018; Choe and Kim, 2019l; Chi et al., 2020), which were conducted in a similar way to our scale development process (a scale development study consisting of several investigations and analyzes). Thanks for your understanding.
References:
Kim, E., Tang, L. R., & Bosselman, R. (2018). Measuring customer perceptions of restaurant innovativeness: Developing and validating a scale. International Journal of Hospitality Management, 74, 85-98.
Choe, J. Y. J., & Kim, S. S. (2019). Development and validation of a multidimensional tourist’s local food consumption value (TLFCV) scale. International journal of hospitality management, 77, 245-259.
Chi, C. G. Q., Chi, O. H., & Ouyang, Z. (2020). Wellness hotel: Conceptualization, scale development, and validation. International Journal of Hospitality Management, 89, 102404.
The logical structure of the introduction chapter does not really come through either. At first, the presentation of the market trends and the situation caused by the COVID epidemic and thus the penetration of the online market is understandable, but the content between lines 39 and 44 is for me disjointed and not comprehensible. In lines 45-46 the authors write: 'there has been a great deal of research related to service quality in the context of online food purchases' - not a single one is cited. In the literature review, I felt there were several areas where references were lacking, and I would rework this chapter in this respect.
â–¶ Thanks for your valuable feedback.
As you mentioned, lines 39-44 is somewhat disjointed, so the authors deleted this part and revised the introduction overall as follows on page 1
(on page 1)
Due to time savings and convenience, several experts predict that the growth of the fresh food delivery platform business will continue even after pandemic is over [4, 5, 6]. Amazon’s global sales in the food category are predicted to increase to $26.7 billion by 2026 from $14.5 billion in 2021. Amazon has been investing in its online grocery business, and the Amazon Fresh service has been expanding in international markets as well [4]. Further, according to a report by McKinsey & Company (2021) [5], approximately 70% of respondents’ preferences for fresh food shopping online changed after the COVID 19 outbreak. Therefore, the fresh food delivery platform would be a new business model and distribution channel to food service industry, and investigation of platform service would be necessary to promote sustainability in the foodservice industry sector.
The lines 45-46 were deleted during the revision process of the paper. Additionally, the authors corrected this paper as follows.
example.
(on page 2)
A fresh food delivery platform is an e-commerce that specializing in the fresh food category [16]. This platform sells food products, fresh foods (e.g., vegetables, fruits, meat, and fish), refrigerated and frozen foods, and other groceries, and offers next-day door to door delivery service [8].
(on page 4)
Since fresh food is a product that easily deteriorates due to time and temperature, product quality has been suggested as a factor that affects consumer purchases [25, 37]. Additionally, financial benefits (e.g., discounts and free gifts) [25, 28, 30, 34] and website design [23, 27, 28, 33] were suggested as factors in the literature.
(on page 4)
Many researchers have investigated service quality attributes and suggested theoretical models to examine the relationship between these attributes and various behavioral variables[23, 24, 25, 26, 30, 33, 34].
Material and Methods chapter:
I do not consider the description of study 1 to be sufficiently thorough. The description of the Big Data analysis is acceptable, but the expert interview and the literature review are particularly weak. In the description of the expert interview, they start by analysing literature and then cite 6 articles. This gives the impression that 6 articles were considered as part of the literature review when constructing the scale. But they also did interviews on the side. This is not good. Besides, very little is revealed about the interviews.
â–¶ Thanks for your comment.
The authors added more detail information about study 1 in the main document. Author added “literature review” phase in the part of the Study 1 (page 5).
The authors also described the contents of the interview additionally on page 7 as follows.
(on page 5)
- Scale development of the fresh food delivery platform service quality
4.1. Study 1: Scale development (Item generation)
The purpose of the Study 1 was to identify the elements that constitute the service quality of fresh food delivery platforms. Based on the existing literature, the data from text-mining, and expert interviews, as many items as possible were generated that rep-resent service quality.
4.1.1. Step 1: Literature review
The first step in scale development was to review existing literature and figure out various service quality dimensions of the fresh food delivery platform and generate items. In order to identify what service quality attributes the fresh food delivery plat-form should have, previous literature related with consumer behavior studies in the field of e- commerce dealing with fresh food and food items [23, 24, 25, 26, 27, 28, 29, 30, 32, 33, 34] were thoroughly reviewed. Based on the existing literature, service quality attributes related to information quality[23, 27], price [23, 28, 29], product assortment [30], problem resolution [24, 25, 32], delivery quality [25, 32, 34], ease of use [24, 25, 26], trendiness [35, 36], sales promotion [25, 28, 30, 34]), product quality [24, 25, 33], design [23, 27, 28] were revealed and related specific measurement items were organized for item generation.
(on page 7)
Interviewees were asked about service attributes, business experiences, and their opinions of fresh food delivery platform services. For instance, the following questions were asked: “What are the service attributes of fresh food delivery platform services?”; “What do you consider important when selling food products including fresh food online?”; and “What do you think customers care about when buying fresh food online?” In the interview, several significant keywords, such as ‘small portion or variety of portion sizes’, ‘next day delivery’, ‘delivery damage control’, ‘temperature-controlled delivery’, ‘delivery accuracy’, ‘accurate product information’, and ‘eco-friendly packaging’, were mentioned by experts.
The results chapter should be consolidated in terms of the presentation of indicators. Some indicators are simply described as such (KMO and Bartlett's test), while others are nicely explained as to whether a given value is appropriate or not.
â–¶ Thanks for your advice.
The authors explained the adequate value of KMO and Bartlett’s test as follows.
(on page 8) When KMO≥0.9 is considered very appropriate; 0.8<KMO≤0.9 represents fairly appropriate; 0.7<KMO≤0.8 represents appropriate; 0.6<KMO≤0.7 represents barely appropriate; 0.5<KMO≤0.6 is considered inappropriate. The KMO measure of sampling adequacy was 0.953 and Bartlett’s test of Sphericity was 16476.095 (p<0.001), indicating the appropriateness of factor analysis
It was confusing to me that the material and methods chapter does not yet focus on the material and methods issues of the third study, but does within the results, and then there is a results chapter within the results. I suggest that what is material and method should be in that chapter and what is results should be in the results chapter.
â–¶ Thank you for comments. For better understanding of this research, we added the analysis process briefly on page 12
(on page 12)
4.3.2. Data analysis
The reliability is judged through the Cronbach’s alpha value obtained using SPSS version 18. In addition, CFA was conducted to evaluate construct validity, convergent validity, and discriminant validity using AMOS version 18. The sample was divided into two groups and cross-validation was checked by performing a Chi-square difference test. Additionally, nomological validity was confirmed through correlation analysis between factors.
However, the strength of the article is the discussion and further research chapters.
â–¶ Thanks for your positive feedback. Thank you very much for the reviewer’s valuable comments for the improvement of this manuscript.

Reviewer 3 Report
In this study, the researchers developed a scale for measuring the service quality of fresh food delivery platforms. While this is an interesting topic, there are a number of significant issues throughout the manuscript. I have listed my biggest concerns below:
Based on the introduction as it stands, it is unclear to me why this scale was necessary to create. It would be helpful if you could talk about current trends in online food shopping in different parts of the world based on where we are in the pandemic. Are people still purchasing perishable items online now that many countries are encouraging more regular, pre-COVID habits? You say that “obviously consumer willingness has increased” and that “online shopping trendwill [typo?] endure post-pandemic”, but I’m not sure these statements are backed up by data (you don’t include citations).
It is also unclear to me what you mean by “service quality” in the introduction, and what the “service quality dimensions and attributes” of fresh food delivery platforms are that necessitate the creation of a new scale beyond the ones used by other fields like online shopping. Can you define “service quality” in the introduction, and provide examples of unique dimensions and attributes of service quality for fresh food delivery that are needed?
The literature review section contains some very useful information (like the definition of a fresh food delivery platform) that would be helpful to include in the introduction.
Moving the methods section before the literature review section would be helpful so that you could clarify what type of literature review you conducted, and how you went about doing it. Was it systematic, scoping, or narrative? How did you choose search terms, and narrow down the results? Which exact topics did the literature review cover, and did you narrow the research down by which country it was conducted in? As of now, the literature review section feels like an expanded introduction/background section, but it’s difficult to follow because I don’t understand how it was structured.
In the methods, you use the term “big data analysis” multiple times, but can you be more specific about what you mean? What types of “documents” related to fresh food delivery platform services did your data consist of? Research articles, business reports, other things?
How were the 6 experts chosen for this study? Can you provide justification for why people who work in the foodservice industry are experts on fresh food delivery platforms?
Can you provide more detail about the surveys you distributed? Which online survey company did you use to distribute them? Besides having purchased food products through a fresh food delivery platform within the last month, were there any other inclusion/exclusion criteria? Did participants have to live in a specific country? Were they provided an incentive to participate?
Why does the Food PlatQual scale include an item about whether the platform offers a variety of processed foods? Isn’t this a scale used to assess fresh food delivery platforms?
In the limitations section you say that items related to "product quality, such as freshness of food, taste, and safety were refined in the analysis process and ultimately not included in Food PlatQual." Why weren’t they included? These attributes seem key to measure when thinking about the quality of a fresh food delivery platform.
Author Response
Thanks for taking your valuable time to review this article.
The author have tried to improve the quality of the paper based on your advice.
In this study, the researchers developed a scale for measuring the service quality of fresh food delivery platforms. While this is an interesting topic, there are a number of significant issues throughout the manuscript. I have listed my biggest concerns below:
Based on the introduction as it stands, it is unclear to me why this scale was necessary to create. It would be helpful if you could talk about current trends in online food shopping in different parts of the world based on where we are in the pandemic. Are people still purchasing perishable items online now that many countries are encouraging more regular, pre-COVID habits? You say that “obviously consumer willingness has increased” and that “online shopping trendwill [typo?] endure post-pandemic”, but I’m not sure these statements are backed up by data (you don’t include citations).
â–¶ Thanks for your useful comments. The fresh food online commerce will continue to grow after COVID-19, and consumers' preference for online purchases of fresh food will also increase. Fresh food has different characteristics (short shelf life, freshness, corruption, and purchase frequency, etc.) from the general factory system. Therefore, the needs that consumers expect from the fresh food platform will be different from other e-commerce. However, systematically developed measurement scales for the fresh food platform have not developed yet. Scale development is necessary because measures specialized in business and product will provide more specific implications and insights. The authors added the necessity of developing this scale. Revised introduction of the article as below on page 2.
(on page 2)
With the growth of the fresh food delivery platform business, the service companies have faced intense competition. Thus, the importance of service quality management for customer satisfaction and sustainable growth is increasing [9]. Service quality of fresh food delivery platform could be defined as the degree of difference between the service that consumer expect from the fresh food delivery platform and consumer’s perceived performance that they experience when purchasing items through services [10]. Service quality is closely related to success of business and customer satisfaction [11]. For successful business, companies should identify which service quality attributes their services possess and then determine how to manage these attributes to meet consumers’ expectations [12]. Unfortunately, the service quality dimensions and attributes of fresh food delivery platforms have not been thoroughly investigated, and service quality measure has not been developed through a systematic process. Thus, the majority of research of online fresh food purchasing behavior derived their service quality factors by referring to previous studies in similar fields, such as online shopping or other types of online services, rather than applying systematically developed measurement items that reflect the unique features of fresh food delivery platforms. Unlike manufactures, fresh food and food products are in the nature of perishable and temperature sensitive items [13]. Besides, fresh food shopping is more frequent activity than shopping for other product category and a regular and routine tasks in the life [14]. Therefore, the factors such as delivery service, temperature-related factors, easy order process, and easy return are should be reflected to consist service quality dimensions of fresh food delivery platform. An industry-specific instrument that considers the unique characteristics of these market offerings should be developed to evaluate service quality and obtain clearer insights and implications.
â–¶Consumer behavior, including food purchase affected by the pandemic, will continue even after the pandemic, so consumers who perceive the many benefits of convenience and time-saving online channels will continue to purchase fresh food online(progressivegrocer, 2021).
Reference :
Progressivegrocer. The Post-Pandemic Fresh Food Consumer, https://progressivegrocer.com/post-pandemic-fresh-food-consumer
(on page 1)
Due to time savings and convenience, several experts predict that the growth of the fresh food delivery platform business will continue even after pandemic is over [4, 5, 6]. Amazon’s global sales in the food category are predicted to increase to $26.7 billion by 2026 from $14.5 billion in 2021. Amazon has been investing in its online grocery business, and the Amazon Fresh service has been expanding in international markets as well [4]. Further, according to a report by McKinsey & Company (2021) [5], approximately 70% of respondents’ preferences for fresh food shopping online changed after the COVID 19 outbreak.
It is also unclear to me what you mean by “service quality” in the introduction, and what the “service quality dimensions and attributes” of fresh food delivery platforms are that necessitate the creation of a new scale beyond the ones used by other fields like online shopping. Can you define “service quality” in the introduction, and provide examples of unique dimensions and attributes of service quality for fresh food delivery that are needed?
â–¶ Thanks for your feedback.
The authors stated the definition of service quality in the introduction and re-organized the part explaining the service quality. In addition, authors emphasized the necessity of the scale development specialized in the fresh food delivery platform.
(on page 2)
With the growth of the fresh food delivery platform business, the service companies have faced intense competition. Thus, the importance of service quality management for customer satisfaction and sustainable growth is increasing [9]. Service quality of fresh food delivery platform could be defined as the degree of difference between the service that consumer expect from the fresh food delivery platform and consumer’s perceived performance that they experience when purchasing items through services [10]. Service quality is closely related to success of business and customer satisfaction [11]. For successful business, companies should identify which service quality attributes their services possess and then determine how to manage these attributes to meet consumers’ expectations [12]. Unfortunately, the service quality dimensions and attributes of fresh food delivery platforms have not been thoroughly investigated, and service quality measure has not been developed through a systematic process. Thus, the majority of research of online fresh food purchasing behavior derived their service quality factors by referring to previous studies in similar fields, such as online shopping or other types of online services, rather than applying systematically developed measurement items that reflect the unique features of fresh food delivery platforms. Unlike manufactures, fresh food and food products are in the nature of perishable and temperature sensitive items [13]. Besides, fresh food shopping is more frequent activity than shopping for other product category and a regular and routine tasks in the life [14]. Therefore, the factors such as delivery service, temperature-related factors, easy order process, and easy return are should be reflected to consist service quality dimensions of fresh food delivery platform. An industrytrum-specific instrument that considers the unique characteristics of these market offerings should be developed to evaluate service quality and obtain clearer in-sights and implications.
The literature review section contains some very useful information (like the definition of a fresh food delivery platform) that would be helpful to include in the introduction.
â–¶ Thanks for your comment.
As your suggestion, the authors added clear definition of the platform to the introduction as 'online commerce that sells various foods including fresh foods and delivers them to your home'. Please see page 2.
(on page 2)
Therefore, this study aimed to fill this critical research gap by developing a scale to assess the service quality of fresh food delivery platforms. In this study, we defined the fresh food delivery platform as an online commerce that sells a variety of food products including fresh food and delivers them to consumer’s home. This research employed the systematic process of instrument development outlined by Churchill (1979) [15].
Moving the methods section before the literature review section would be helpful so that you could clarify what type of literature review you conducted, and how you went about doing it. Was it systematic, scoping, or narrative? How did you choose search terms, and narrow down the results? Which exact topics did the literature review cover, and did you narrow the research down by which country it was conducted in? As of now, the literature review section feels like an expanded introduction/background section, but it’s difficult to follow because I don’t understand how it was structured.
â–¶ Thank you for your valuable feedback.
First, the authors collected and reviewed a wide range of previous studies related to online grocery shopping, online food shopping, and online fresh food purchase, then the authors delineated the properties of the online fresh food platform. Since there were not many studies related to online food fresh food purchase, the country of the research background was not considered.
Second, a ‘literature review step’ was added to the methodology section based on your suggestions. (Methodology part of the first version was consisted of big data and expert interviews, but for a better understanding of this paper, it was re-composed of literature reviews, big data, and expert interviews.)
Please see page 5 and 6.
(on page 5~6)
4.1. Study 1: Scale development (Item generation)
The purpose of the Study 1 was to identify the elements that constitute the service quality of fresh food delivery platforms. Based on the existing literature, the data from text-mining, and expert interviews, items were generated that represent service quality.
4.1.1. Step 1: Literature review
The first step in scale development was to review existing literature and figure out various service quality dimensions of the fresh food delivery platform and generate items. In order to identify what service quality attributes the fresh food delivery plat-form should have, previous literature related with consumer behavior studies in the field of e- commerce dealing with fresh food and food items [23, 24, 25, 26, 27, 28, 29, 30, 32, 33, 34] were thoroughly reviewed. Based on the existing literature, service quality attributes related to information quality[23, 27], price [23, 28, 29], product assortment [30], problem resolution [24, 25, 32], delivery quality [25, 32, 34], ease of use [24, 25, 26], trendiness [35, 36], sales promotion [25, 28, 30, 34]), product quality [24, 25, 33], design [23, 27, 28] were revealed and related specific measurement items were organized for item generation.
4.1.2. Step 2: Big data analysis
The second step in scale development was big data analysis to explore consumers’ perceptions of fresh food delivery platforms. Big data can provide a wide range of consumer insights into research areas that conventional research techniques have not identified well [41]…………………….
In the methods, you use the term “big data analysis” multiple times, but can you be more specific about what you mean? What types of “documents” related to fresh food delivery platform services did your data consist of? Research articles, business reports, other things?
â–¶ Thanks for your comment.
The data used for big data are drawn from blogs posted on 'Naver' and 'Daum', two of Korea's leading online portal sites. In order to reflect the opinions of consumers on the food distribution platform, we collected data from blogs created by consumers rather than articles and business reports. In this regard, additional explanations have been added to page 6.
(on page 6)
“Naver” is definitely the largest portal site in South Korea. According to the ICT ministry data, the average daily user number of “Naver” for the last three months of 2021 is 40.3 million, which is about 78% of the total number of Koreans [43]. “Naver” offers various IT services, such as search engines, blog, communities, mail, and more [44]. “Daum” is Korea’s second-largest portal sites and it also provides similar services like “Naver” [45].
How were the 6 experts chosen for this study? Can you provide justification for why people who work in the foodservice industry are experts on fresh food delivery platforms?
â–¶ In order to conduct this study, the authors wanted to invite experts who majored in restaurant management and who have worked in the food industry especially in online fresh food commerce for at least 5 years to participate in the interview. For this purpose, among those who graduated from Kyung Hee University or Kyung Hee graduate school, six people who qualified above conditions were contacted and interviewed, and incentives were paid. Two experts are working on an online salad delivery platform, and two experts are in charge of online distribution and sales management of fresh food and food products in the foodservice company. One expert develops refrigerated meal kits and sells them online. The other one is a university lecturer with experience working on a fresh food delivery platform. They have special expertise in the fresh food and food products where freshness and temperature control are important. So they are well aware of trends in food distribution business and consumer behavior and have ability to provide useful and practical information for this research.
The authors described more detail process of experts’ interview in the document on page 6 and 7.
(on page 6 and 7)
Can you provide more detail about the surveys you distributed? Which online survey company did you use to distribute them? Besides having purchased food products through a fresh food delivery platform within the last month, were there any other inclusion/exclusion criteria? Did participants have to live in a specific country? Were they provided an incentive to participate?
â–¶ To collect the data for consumer survey, the authors contacted the online research company, Macromil Embrain. This company is the No.1 online research agency with the largest consumer panel (3million) in Asia. In addition, it has the largest panel (1.5 million) in Korea. Macromil Embrain distributed questionnaires to their panels. With the screening items, this company extracted qualified panels for the research purpose. there were no inclusion/exclusion criteria. Consumers who had purchased a variety of products, including fresh food, from a Korean food distribution platform within the last month were allowed to participate in the survey. We did not put any restrictions on where we live, but since the survey was conducted in Korea. The participants in the survey were consumers living in Korea. Incentives were provided to those who took part in the survey.
For a better understanding of this research, the authors wrote detailed information about the online research company and sampling process in the main document as follows on page 7. Thanks for your useful feedback.
(on page 7)
The consumer survey was conducted in South Korea by Macromil Embrain which is the No.1 online research agency with the largest panel (1.5 million) in South Korea [49]. The questionnaire contained the 55 items from Study 1. Each item was measured using a 7-point Likert scale anchored by 1 (strongly disagree) and 7 (strongly agree). Questionnaires were distributed by an online survey company, to randomly selected respondents who had bought food products through Korean fresh food delivery platform (e.g., Market Kurly, Oasis, Coupang Fresh, and etc.) within the last month. The subjects of this survey are the top 6 down delivery service providers in Korea considering the number of users [50]. These platforms specialize in selling fresh food and food products and provide services that allow consumers receive orders the next day immediately. “Market Kurly” is a pioneer of the fresh food delivery service and had more than 10 million users at the end of 2021 [51, 52]. And “Coupang Fresh” is fastest growing online fresh food commerce in South Korea [52].
In the sampling process, in order to determine whether or not they have actually purchased food through a food distribution company, they were asked to select the buying channel they have used for food purchase within the last 1 month. And the respondents who selected online shopping mall among various items such as offline market, TV home shopping channel, and online were extracted. Next, after asking what kind of online shopping mall respondents used to purchase food within 1 month, the respondents who chose to use the online shopping mall as defined in this study were finally selected and participated in the survey. A total of 2,079 respondents to participate the consumer survey but 1,446 unqualified responses were screened out based on screening items that asked which types of online shopping sites respondents had used before. 65 insincere cases were also eliminated.
Why does the Food PlatQual scale include an item about whether the platform offers a variety of processed foods? Isn’t this a scale used to assess fresh food delivery platforms?
â–¶ This is the author's translation error. The "processed food" mentioned by the authors refers to refrigerated and frozen foods (e.g., "frozen fruit", "frozen fish", and "frozen meat"). The authors apologized for the confusion caused by the translation error. The word was properly corrected. The fresh food distribution platform not only sells fresh food but also sells and delivers refrigerated and frozen food that needs temperature control. When consumers buy fresh food on the platform, they also buy various categories of products together. Therefore, it is believed that a group of various products can affect the consumer's choice of choosing one of the fresh food delivery platforms among the various platforms.
In the limitations section you say that items related to "product quality, such as freshness of food, taste, and safety were refined in the analysis process and ultimately not included in Food PlatQual." Why weren’t they included? These attributes seem key to measure when thinking about the quality of a fresh food delivery platform.
â–¶ Thanks for pointing out this issue.
The authors thought that this is because consumers recognize that delivery quality is more important than product quality when using fresh food delivery platforms. This point might imply that consumer needs might be different when using an offline store and when using an online store. The quality of fresh food is greatly affected by delivery without a damage, temperature-controlled delivery, and delivery time. Therefore, it seems that product quality may have already been reflected in the delivery quality dimension. However, as you mentioned, product quality is a factor that should not be overlooked in consumer behavior, we mentioned the need of additional research for re-examination of product quality in the Limitations and Future Research section on page 16.
(on page 16)
Second, this study thoroughly identified service quality attributes based on qualitative and quantitative methods, but items related to product quality, such as, freshness of food, taste, and safety were deleted in the analysis process and not included in Food PlatQual. Since the freshness and safety of food are closely related to delivery quality such as delivery without a damage, delivery temperature, and delivery time, it is presumed that some of them were reflected in the delivery quality dimension. Moreover, it is thought that this is because consumers recognize that delivery quality is more important than product quality when using fresh food delivery platforms. This point might imply that consumer needs when using an offline store and when using an online store may be different. In addition, the fresh food delivery platforms are playing the role of a retailer rather than a producer of fresh food or food products on a food distribution platform. Therefore, it is assumed that the items of product quality had refined from the service quality items that the fresh food delivery platform should have. However, since the product quality is still an important part in overall e-commerce, it is necessary to re-examine these items in future studies.

Round 2
Reviewer 2 Report
Overall, I think the article has improved a lot. The logical structure has become much easier to follow thanks to the additions.
I just have one minor comment:
In the second table, the end of SQ 1-4 sentence is missing.
Author Response
Thanks for your comment.
We revised SQ 1-4 sentence in the Table 2 as below:
“The platform offers detailed product information.”
Reviewer 3 Report
Thank you for your thoughtful edits and responses. A few more thoughts:
Pages 4-5, Figure 1: In the text, you say that big data analysis was the first step in your process, followed by a second step involving a literature review. In Figure 1, however, it looks like the first step involves the literature review, big data analysis, and expert interviews. Can you clarify?
Pages 5-6: I appreciate that you added a methods step about your literature review. Could you please add just a little more detail about how you reviewed previous literature? What databases did you search, and what search terms did you use?
Page 6, Lines 265-266: You open the “Expert’s interview” section by talking about the literature review, but I think you can get rid of this first sentence and just start by saying that you conducted in-depth interviews with six experts in July 2021.
Page 7, lines 290-295: Can you clarify whether the six experts you’re referring to here are the six experts from the interviews, or different experts?
Page 7, Data collection and sampling: Can you add a little more information about the online survey company that you used, and what the inclusion criteria was for respondents? Did the survey company only recruit participants >18 years old located in South Korea who had purchased food from 1+ of 6 Korean fresh food delivery platforms in the last month? Or were there other inclusion criteria? Can you also clarify what the top 6 delivery platforms were that you included in your inclusion criteria? In Table 1 it looks like all participants reported shopping at only 4 delivery platforms.
Page 7, lines 326-329: I think there’s a typo in here—can you clarify what you mean by “A total of 2,079 respondents to participate in the consumer survey” means? Were 2,079 participants invited to take the survey, or did 2,079 participants actually try to complete the survey? Do you know how many participants were invited to take the survey by the survey company, and whether they received any incentives (like money) to take part?
Page 7, lines 328-329: Can you clarify what “insincere cases” means? Did you have an attention check question or some other quality control that these people didn’t pass?
Page 7, lines 331-332: This is a small issue, but the word “majority” should be changed to “plurality” here, because 39.1% of respondents is not a majority.
Page 9, Table 2: I think there’s a typo or missing information for SQ 1-4 : “The platform offers detailed”. Can you also write out what “SNS” stands for (SQ 4-4) or include a note about what the abbreviation means in the footnote of the table?
Page 10, line 388: do you mean Study 2 here instead of Study 3?
Page 13, Cross validity: Can you justify why you only tested for cross validity between those two age groups and between two random samples, vs. between different age groups, education levels, gender, etc?
Author Response
Thanks for your comments.
We really appreciate your valuable advice for our research.
We tried to reflect your comments to this article.
Thank you for your thoughtful edits and responses. A few more thoughts:
Pages 4-5, Figure 1: In the text, you say that big data analysis was the first step in your process, followed by a second step involving a literature review. In Figure 1, however, it looks like the first step involves the literature review, big data analysis, and expert interviews. Can you clarify?
Thanks for your valuable comment.
The process of Study 1 was not presented consistently. Therefore, author revised the process of Study 1 in a consistent order of literature review, big data analysis, and expert interview in the main document (page 5).
(on page 5)
This study developed potential items based on three sources: research literature, big data, and expert interviews. First, previous literature related to online food purchase was reviewed. Subsequently, big data analysis was performed to ensure that consumers’ broad opinions regarding fresh food delivery platforms were reflected in the development of the scale. Expert’s interview was carried out to examine the initial pool of items. Then, an online survey was conducted to purify the scales and assess the initial validity.
Pages 5-6: I appreciate that you added a methods step about your literature review. Could you please add just a little more detail about how you reviewed previous literature? What databases did you search, and what search terms did you use?
The authors added more detailed information of literature review. The search keywords used for reviewing previous studies are presented on page 5.
(on page 5)
The first step in scale development was to review existing literature and figure out various service quality dimensions of the fresh food delivery platform and generate items. Prior studies were reviewed using keywords such as online fresh food purchase, online food shopping, online grocery shopping, and etc. In order to identify what service quality attributes the fresh food delivery platform should have, previous literature related with consumer behavior studies in the field of e- commerce dealing with fresh food and food items [23, 24, 25, 26, 27, 28, 29, 30, 32, 33, 34] were thoroughly reviewed.
Page 6, Lines 265-266: You open the “Expert’s interview” section by talking about the literature review, but I think you can get rid of this first sentence and just start by saying that you conducted in-depth interviews with six experts in July 2021.
Thanks for your comment.
As you mentioned, we got rid of the first sentence as below.
(on page 6)
4.1.3. Step 3: Expert’s interview
To generate items for Food PlatQual, relevant previous studies on online grocery shopping [23, 24, 27, 31, 32, 33] were reviewed. At the same time, The in-depth interviews were conducted with six experts during the period of July 1-July 10, 2021. In order to conduct this study, we wanted to invite experts who majored in restaurant management and who have worked in the food industry especially in online fresh food commerce for at least 5 years to participate in the interview.
Page 7, lines 290-295: Can you clarify whether the six experts you’re referring to here are the six experts from the interviews, or different experts?
All six experts interviewed in the previous step were also participated in the survey. For better understanding, we clarified this point in the manuscript as followz.
(on page 7)
Six experts who participated in the expert’s interview judged the 69 attributes on a five-point Likert scale anchored by 1 (strongly disagree) and 5 (strongly agree). 14 items with an average score lower than 4 (agree) were removed, per Brakus et al.’s (2009) [48] study, and a final set of 55 items remained.
Page 7, Data collection and sampling: Can you add a little more information about the online survey company that you used, and what the inclusion criteria was for respondents? Did the survey company only recruit participants >18 years old located in South Korea who had purchased food from 1+ of 6 Korean fresh food delivery platforms in the last month? Or were there other inclusion criteria? Can you also clarify what the top 6 delivery platforms were that you included in your inclusion criteria? In Table 1 it looks like all participants reported shopping at only 4 delivery platforms.
The survey participants were over 18 years old who had purchased fresh food from 6 major companies. Most of the respondents answered that they used 4 major companies (Marketkurly, Coupang fresh, Oasis, and Hello nature) in Korea, and the respondents who have used the other 2 services (SSG and To home) were few, so they were all included in the ‘others’ group.
As your comment, we mentioned about it in Table 1 as follows.
Page 7, lines 326-329: I think there’s a typo in here—can you clarify what you mean by “A total of 2,079 respondents to participate in the consumer survey” means? Were 2,079 participants invited to take the survey, or did 2,079 participants actually try to complete the survey? Do you know how many participants were invited to take the survey by the survey company, and whether they received any incentives (like money) to take part?
A total of 2,079 panels actually tried to participate the survey but 1,446 responses were not qualified for this study (e.g., no experience of purchasing food through fresh food delivery platforms). Initially, the survey company distributed e-mail (invitation to survey) to 8,063 panels but, only 2,079 accessed the survey. And participants who completed the survey received a monetary reward from the survey company.
As your advice, author clarified the meaning of that sentence as below.
(on page 7)
A total of 2,079 respondents to participate the consumer survey but 1,446 unqualified responses.
à A total of 2,079 respondents tried to participate the consumer survey but 1,446 unqualified responses
Page 7, lines 328-329: Can you clarify what “insincere cases” means? Did you have an attention check question or some other quality control that these people didn’t pass?
The responses repeatedly checked the same number on a Likert scale regarded as insincere cases and were eliminated by the survey company (For example, responses that checked all questions in 4(neutral), and etc.). The survey company refined these cases and 550 responses handed over to authors for analysis.
Page 7, lines 331-332: This is a small issue, but the word “majority” should be changed to “plurality” here, because 39.1% of respondents is not a majority.
Thanks for your useful comment. Authors corrected this sentence as follows.
(on page 8)
A majority of respondents were 30-39 years old (39.1%), ~~ à A plurality of respondents were 30~39 years old (39.1%), ~~~
Page 9, Table 2: I think there’s a typo or missing information for SQ 1-4 : “The platform offers detailed”. Can you also write out what “SNS” stands for (SQ 4-4) or include a note about what the abbreviation means in the footnote of the table?
Thanks for your comment. We revised the sentence in Table 2 as below.
SQ 1-4. The platform offers detailed product information.
SQ 4-4. The platform communicates with customers through social networking services.
Page 10, line 388: do you mean Study 2 here instead of Study 3?
Do you mean this sentence? “To confirm the 25 Food PlatQual items discovered in Study 2, a second consumer survey was carried out in Study 3.”?
The authors think we wrote it right way. This sentence means that Study 3 (a second consumer survey) was carried out to confirmed the items derived from Study 2.
Page 13, Cross validity: Can you justify why you only tested for cross validity between those two age groups and between two random samples, vs. between different age groups, education levels, gender, etc?
To determine cross-validity, it is necessary to analyze the difference between the two groups. For analysis, the sample sizes of the two groups needed to be similar, but it was not easy to assign similar sample sizes when classifying them based on gender, educational background, etc. Therefore, we divided two groups according to the respondent’s age. Thanks for your understanding.